# Catalytically efficient Ni-NiO$_x$-Y$_2$O$_3$ interface for medium temperature water-gas shift reaction

Kai Xu [1], Chao Ma [2], Han Yan[1], Hao Gu[3], Wei-Wei Wang [1], Shan-Qing Li [4], Qing-Lu Meng [1], Wei-Peng Shao [1], Guo-Heng Ding[1], Feng Ryan Wang [3✉] & Chun-Jiang Jia [1✉]

The metal-support interfaces between metals and oxide supports have long been studied in catalytic applications, thanks to their significance in structural stability and efficient catalytic activity. The metal-rare earth oxide interface is particularly interesting because these early transition cations have high electrophilicity, and therefore good binding strength with Lewis basic molecules, such as H$_2$O. Based on this feature, here we design a highly efficient composite Ni-Y$_2$O$_3$ catalyst, which forms abundant active Ni-NiO$_x$-Y$_2$O$_3$ interfaces under the water-gas shift (WGS) reaction condition, achieving 140.6 μmol$_{CO}$ g$_{cat}^{-1}$ s$^{-1}$ rate at 300 °C, which is the highest activity for Ni-based catalysts. A combination of theory and ex/in situ experimental study suggests that Y$_2$O$_3$ helps H$_2$O dissociation at the Ni-NiO$_x$-Y$_2$O$_3$ interfaces, promoting this rate limiting step in the WGS reaction. Construction of such new interfacial structure for molecules activation holds great promise in many catalytic systems.

[1] Key Laboratory for Colloid and Interface Chemistry, Key Laboratory of Special Aggregated Materials, School of Chemistry and Chemical Engineering, Shandong University, Jinan 250100, China. [2] College of Materials Science and Engineering, Hunan University, Changsha 410082, China. [3] Department of Chemical Engineering, University College London, Roberts Building, Torrington Place, London WC1E 7JE, UK. [4] Key Laboratory of Micro-Nano Powder and Advanced Energy Materials of Anhui Higher Education Institutes, Chizhou University, Chizhou 247000, China. ✉email: ryan.wang@ucl.ac.uk; jiacj@sdu.edu.cn

**P**hysicochemical interfaces between the metal and support are important in heterogeneous catalysis[1]. Construction of catalytically active interfaces between metals and supports has attracted extensive attentions. The metal support interaction leads to high dispersion[2,3] or even over-encapsulation of active metals[4]. Therefore, it is crucial to design and control suitable metal-support interaction and thus create abundant and effective interfacial active sites for molecule transformation. In particular, the metal interaction with third row (MgO[5], Al$_2$O$_3$[6] and SiO$_2$[7]) oxides and fourth row 3d metal oxides (TiO$_2$[1,4,8–12], FeO$_x$[13–15], Co$_3$O$_4$[16,17]) have been widely explored and discussed in terms of electron transfer process[18,19], oxygen vacancies[1,11,12,20] and surface wetting[4]. In comparison, fifth or sixth row oxides, such as rare earth oxides, are seldom used as catalysts support except for CeO$_2$[2,3,20–25]. These early transition rare earth cations are strong Lewis acid and have strong binding to Lewis base such as H$_2$O and NH$_3$. The metal-rare earth oxide interface should have very different properties to those third and fourth row metals. It is a complete surprise that they have been rarely investigated in the H$_2$O or NH$_3$ based reactions, such as water-gas shift (WGS) reaction[11,21–23].

WGS is an irreplaceable reaction to produce H$_2$ from CO and H$_2$O, which is responsible for ammonia synthesis, hydrogenation reactions and hydrogen fuel cells[26]. On one hand, WGS is exothermic and limited by thermodynamic equilibrium at high-temperature (HT). On the other hand, H$_2$O is a very weak oxidant for CO and has very sluggish kinetics at low-temperature (LT). In comparison, medium-temperature (MT) WGS can achieve rapid kinetics with high CO conversion equilibrium[27,28]. Therefore, developing effective and stable catalytic interfaces in the MT region is a practical route to replace the existing cascade HT-LT WGS process. This is usually achieved with Ni-based catalysts[11,12,29–36], which is mostly active between 300 and 400 °C. Existing research focus on CeO$_2$ based support[30–35] mostly due to its abundant oxygen vacancy. However, at MT region, such oxygen vacancy may not be necessary, whereas the surface adsorption and activation of H$_2$O is the key. This is an area that rare earth support, such as Y$_2$O$_3$, has good competitivity. In addition, Y$_2$O$_3$ also offers good thermal stability, high chemical durability and high mechanical strength, promoting the stability of the metal oxide interface.

Here we reported an in situ formed Ni-NiO$_x$-Y$_2$O$_3$ interface during MT-WGS reaction, achieving 140.6 µmol$_{CO}$ g$_{cat}^{-1}$ s$^{-1}$ of CO conversion at 300 °C, which was 4 times to the existing literature reports. Such unique interface was only formed with a 9:1 molar ratio of Ni to Y. The addition of only 10% Y$_2$O$_3$ played a key role in catalytic activity improvement and prevented the catalyst from sintering. In addition, through theoretical simulation we have demonstrated the electron transfer at the interface, which indicated the existence of the interaction intrinsically. The Ni-NiO$_x$-Y$_2$O$_3$ interface was clearly characterized with aberration corrected electron microscopy and in situ Raman spectra. Combined with DFT calculation, H$_2$O molecules dissociated effectively on the Ni-NiO$_x$-Y$_2$O$_3$ interfacial site was proved. And the calculation of the whole reaction process also proved that the Ni-NiO$_x$-Y$_2$O$_3$ interface was very efficient for WGS reaction. The excellent catalytic performance of Ni-NiO$_x$-Y$_2$O$_3$ interface suggested the potential of studying and applying rare earth elements in catalysis, creating new metal-support interface with unconventional surface behaviors.

## Results

### Catalytic performance of the Ni$_9$Y$_1$O$_x$ catalyst in WGS reaction.
Series of Ni$_a$Y$_b$O$_x$ (a: b was the molar ratio of Ni to Y) samples were synthesized via the ultrasonic spray method[21,37,38].

The simultaneous decomposition of Ni and Y precursors in the same sprayed droplet ensured the formation of maximized interface between the two metals (Supplementary Fig. 1). The droplets were sent into a high temperature tube furnace under the purging of N$_2$ to sufficiently undergo pyrolysis reaction, forming the Ni$_a$Y$_b$O$_x$ catalyst. Supplementary Figs. 2, 3a and Fig. 1a showed the activities of Ni$_9$M$_1$O$_x$ (M = Y, Ce, Sm and Al) catalysts and the Ni$_a$Y$_b$O$_x$ catalysts with different Ni/Y ratios for the WGS reaction. The CO conversion of the Ni$_9$Y$_1$O$_x$ catalyst was much higher than other catalysts, reaching more than 90% at 250 °C (pretreatment in the H$_2$ atmosphere, the gas hourly space velocity (GHSV) was 42,000 cm$^3$ g$_{cat}^{-1}$ h$^{-1}$). This catalytic activity transferred the HT to MT WGS reaction for Ni-based catalyst. The different catalytic performance of Ni$_a$Y$_b$O$_x$ catalysts correlated well with their reducing ability (Supplementary Fig. 4). Catalyst with low reduction temperature and large H$_2$ consumption in the H$_2$ temperature-programmed reduction (H$_2$-TPR) profile showed high CO conversion. The Ni$_9$Y$_1$O$_x$ has a small reaction peak at 220 °C and a major one at 400 °C. In comparison, both NiO and Y$_2$O$_3$ have only one feature at 350 and 575 °C, respectively (Supplementary Fig. 5). The peak at 220 °C was therefore related to the Ni-Y$_2$O$_3$ interface. As shown in Fig. 1b, pretreatment in the H$_2$ atmosphere shifted the CO conversion to lower temperature range for almost 150 °C, compared to the air-pretreated one. This result suggested that the oxidation state of the Ni played an important role. The formation of CH$_4$ was only detected at 250 and 300 °C with less than 3% selectivity. Compared to other catalyst system in the literature[11,15,21,30,35,39–41], the Ni$_9$Y$_1$O$_x$ catalyst achieved the highest reaction rate and turnover frequency (TOF) of Ni-based catalysts, even higher than some Cu-based and noble metal catalysts between 250 and 350 °C (Fig. 1c, Supplementary Table 1). It also maintained solid time-on-stream stability at high temperature and with an ultra-high GHSV, in which the conversion decayed by only 2% in 100 h (Fig. 1d). Even in the stream also contained CO$_2$ and H$_2$ (5% CO, 23.3% H$_2$O, 10% H$_2$, 5% CO$_2$, N$_2$), Ni$_9$Y$_1$O$_x$ could still maintain similar catalytic performance comparable to that of commercial Cu-Zn-Al (Supplementary Fig. 3b). The addition of Y$_2$O$_3$ significantly improved the activity and stability of the catalyst, compared to pure Ni sample (with NiO as precursor).

### Structural change of the Ni$_9$Y$_1$O$_x$ catalyst.
The specific surface areas ($S_{BET}$) increased from 12.9 to 55.5 m$^2$ g$^{-1}$ when the Ni content was increased from 5.4% to 67.3% (Supplementary Table 2). Combined with the transmission electron microscope (TEM) and scanning electron microscope (SEM) characterizations (Supplementary Fig. 6), it could be observed that the Ni$_9$Y$_1$O$_x$ catalyst presented spherical morphology (250–1500 nm). Aberration-corrected high-angle annular dark-field scanning transmission electron microscopy (HAADF-STEM) images showed that the sample composed of uniformly dispersed nanoparticles (Fig. 2). The inter-planar spacing of the fresh Ni$_9$Y$_1$O$_x$ (Fig. 2d) sample was the same to the lattice fringe of NiO (111), which was consistent with the X-ray diffraction (XRD) results (Fig. 3a). In addition, no lattice fringe of Y$_2$O$_3$ was observed, indicating that Y$_2$O$_3$ was poorly crystallized due to the low calcination temperature. In comparison, Ni$_9$Y$_1$O$_x$-used (Fig. 2h, i) mainly showed the lattice fringe of metallic Ni, indicating the reduction of Ni$^{2+}$ to Ni$^0$ during the H$_2$ pretreatment. In addition, lattice fringes of Y$_2$O$_3$ were found between Ni particles (Fig. 2h) indicated the partial crystallization of Y$_2$O$_3$. The structural changes of the catalyst were observed more intuitively based on the results of elemental mapping. As shown in Fig. 2e, j, the initial uniform distribution of Ni, Y and O in the fresh Ni$_9$Y$_1$O$_x$ catalyst segregated after reaction. For Ni$_9$Y$_1$O$_x$-used, the Ni particles grew

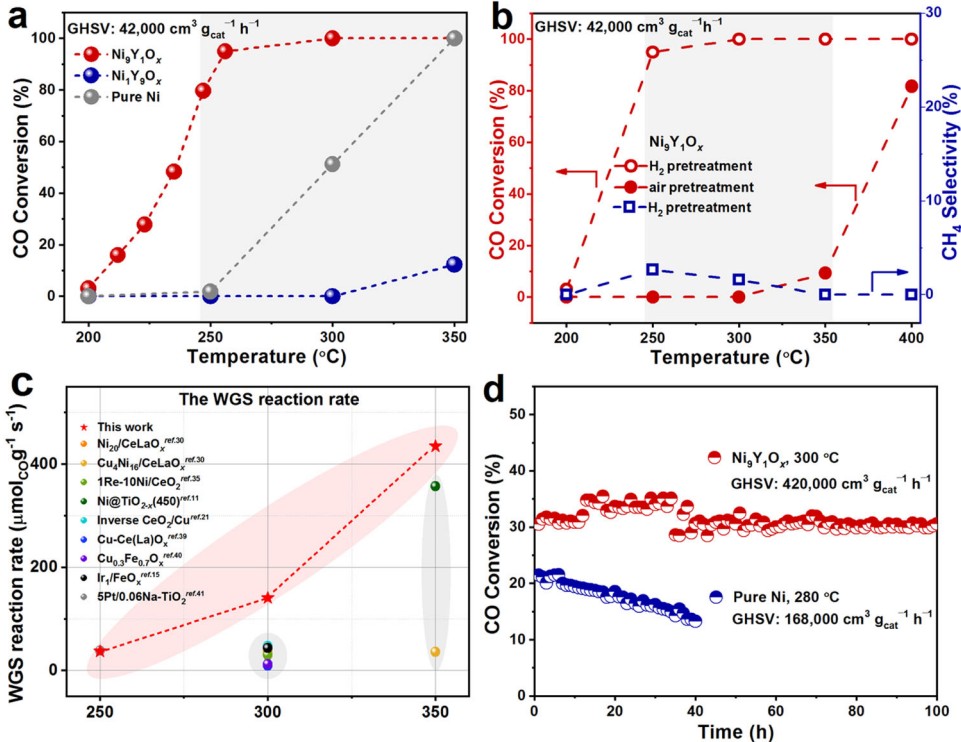

**Fig. 1 Catalytic performance test of the $Ni_aY_bO_x$ catalysts. a** Temperature-dependent activities of the catalysts ($Ni_1Y_9O_x$, $Ni_9Y_1O_x$, pure Ni catalyst, the reaction gas content was 2%CO, 10%$H_2O$, and the rest was $N_2$); **b** temperature-dependent activity and selectivity of $Ni_9Y_1O_x$ under different pretreatment conditions; **c** comparison of reaction rates with other Ni-based catalysts; **d** the long-term stability at a high GHSV ($Ni_9Y_1O_x$: 300 °C, GHSV = 420,000 $cm^3$ $g_{cat}^{-1}$ $h^{-1}$; pure Ni catalyst: 280 °C, GHSV = 168,000 $cm^3$ $g_{cat}^{-1}$ $h^{-1}$).

larger to about 8 nm, and the Y and O were mainly distributed in the surroundings of Ni particles. This structure prevented the further growth of Ni particles and provided abundant interfaces between Ni and $Y_2O_3$. Comparing with TEM images of pure Ni sample (Supplementary Fig. 7), it was clear that $Y_2O_3$ played the crucial role in forming interfaces and stabilizing the catalyst structure.

We further explored the chemical state of the catalyst surface by X-ray photoelectron spectroscopy (XPS). The fitting results of XPS data were integrated in Supplementary Table 3. From Ni 2*p* XPS spectra (Fig. 3b, d), it was seen that the peak of $Ni_9Y_1O_x$-fresh mainly corresponded to $Ni^{2+}$ species, while that of $Ni_9Y_1O_x$-used sample mainly corresponded to $Ni^0$ species[12,30–33]. This observation was consistent well with the observed lattice fringes (Fig. 2h). Trace amount of $Ni^{2+}$ species were detected for $Ni_9Y_1O_x$-used, which was caused by the partial oxidation when exposed in air. From the result of quasi in situ XPS, only $Ni^0$ species were detected on the surface (Fig. 3d). The Y 3*d* (Fig. 3c, e) spectra showed the $Y^{3+}$ species for both samples[42–44]. The main difference was that $Y^{3+}$ in $Y_2O_3$ was mainly detected from the fresh sample; but for the used samples, it was mainly Y carbonate. The catalyst surface change might result from the WGS reaction containing $CO_2$, which was easily absorbed by $Y_2O_3$ on the surface. The bulk $Y^{3+}$ was still in the form of $Y_2O_3$ as confirmed in XRD results (Fig. 3a). The local structure of $Y^{3+}$ was further examined in the Y *K* edge X-ray absorption fine structure (XAFS). At the near edge, pure $Y_2O_3$ crystal showed peaks after the edge (Fig. 3f). Literature suggested that there were two $Y^{3+}$ sites inside $Y_2O_3$ crystal[45]. Site 1 at 17054.4 eV had the same Y-O bonding length, whereas site 2 at 17063.0 eV had different Y-O lengths. Both fresh and used $Ni_9Y_1O_x$ only had the site 1 feature, suggesting weak crystallinity and uniform Y-O bonding length. The near edge spectra also confirmed the $Y^{3+}$ states for both fresh

and used catalysts (Fig. 3f). The Y-O coordination numbers (C.N.) decreased from 7.671 ± 2.033 Å to 5.155 ± 0.392 Å after catalysis (Fig. 3g and Supplementary Table 4). This was associated with the formation of Y-Y coordination at 3.613 ± 0.098 Å. The change in the coordination situation was mainly related to the better crystallization of $Y_2O_3$ after the catalytic reaction, which was consistent with the XRD results (Fig. 3a). The combination of XRD, TEM, XPS and XAFS results suggested that $Y^{3+}$ doped into the NiO lattice at the fresh state. During the reduction with $H_2$, the NiO was reduced to Ni, leaving no space for $Y^{3+}$. Therefore, $Y^{3+}$ was partially converted into crystalline $Y_2O_3$ and formed the Ni-$Y_2O_3$ interface.

Then we explored the nature of the interfacial interactions. Firstly, the theoretical and actual hydrogen consumption of the catalysts was quantitatively calculated according to the $H_2$-TPR result (Supplementary Fig. 4, Supplementary Table 2). The actual $H_2$ consumption was higher than the theoretical value, which indicated that the $Y_2O_3$ species in the catalyst were also reduced due to the formation of the interface. Besides, DFT calculations demonstrated the nature of the interaction was an electrostatic attraction (Supplementary Fig. 8). The charge density calculations suggested that the combination $Y_3O_4$/$NiO_x$/Ni{111} from $Y_3O_4$, $NiO_x$ and Ni{111} appeared noticeable electronic migration ($NiO_x$ gained 1.643 electrons from $Y_3O_4$ and 0.187 electron from Ni{111}).

**The determination of Ni-$NiO_x$-$Y_2O_3$ interfaces.** The phase change of $Ni_9Y_1O_x$ catalyst was further examined with in situ XRD under 5% $H_2$/Ar atmosphere (Fig. 4a). It was found that NiO phase dominated from room temperature to 300 °C. The formation of metallic Ni appeared at 350 °C, which was in good agreement with the ex situ XRD results (Fig. 3a). This also explained the low activity of the air-activated catalyst below

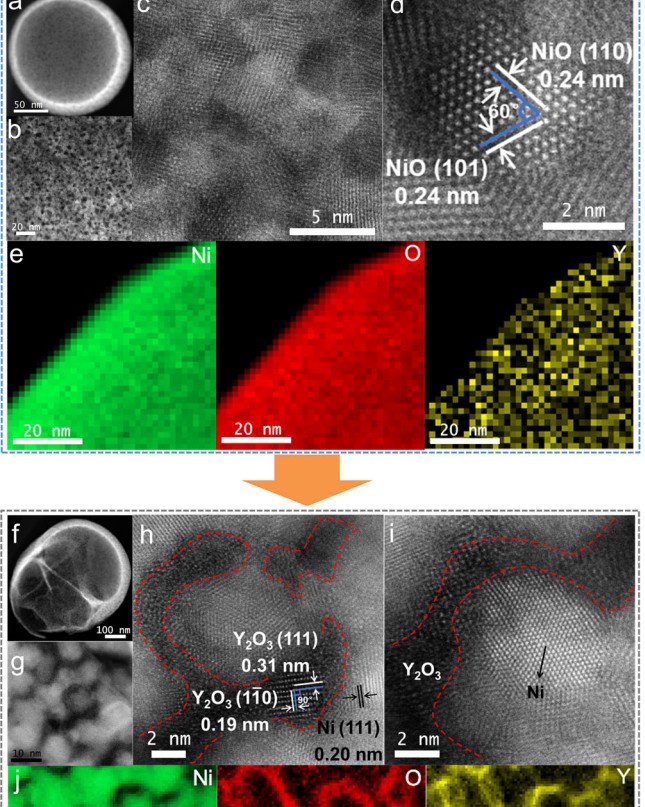

**Fig. 2 The aberration-corrected HAADF-STEM images and elemental mappings.** a–d $Ni_9Y_1O_x$-fresh; f–i $Ni_9Y_1O_x$-used; electron energy-loss spectroscopy (EELS) elemental mapping results of (e) $Ni_9Y_1O_x$-fresh or (j) $Ni_9Y_1O_x$-used.

350 °C because NiO was not reduced below this temperature under reaction condition. In addition, the crystalline dominate size of metallic Ni increased gradually with the temperature rise (Supplementary Table 5), 5.5 nm at 400 °C and 9.6 nm at 500 °C, according to the calculation using Scherrer formula.

We further verified microscopic structure change of the interface during the WGS reaction by in situ Raman spectra. The vibration peak around ~505 $cm^{-1}$ which corresponds to NiO[46–50] was observed in the $Ni_9Y_1O_x$ and pure NiO sample before reaction (Fig. 4b, d, Supplementary Fig. 9). The change of temperature and particle size would cause the shift of NiO vibration peak[50]. In addition, the change of peak position could also be considered as the formation of other Ni-O coordination forms, resulting in the formation of $NiO_x$ ($0 < x < 1$) species. Pretreatment at 400 °C under 5%$H_2$/Ar led to disappearance of NiO vibration peak (Fig. 4b), indicating that NiO was fully reduced to metallic Ni. This observation was consistent with the XRD results (Fig. 4a). Very interestingly, the vibration peak of NiO appeared again in the WGS reaction atmosphere, suggesting the formation of Ni-O bond on the surface. Furthermore, by switching the feeding gas between CO and $H_2O$ (Fig. 4c), it was found that the vibration peak of NiO disappeared when CO was introduced at 400 °C, and then it appeared again after feeding $H_2O$ vapor. This indicated that the O atoms around Ni were

almost fully consumed when the catalyst was treated at high temperature under CO, which was evidenced by the in situ XRD results (Supplementary Fig. 10). The formation of $NiO_x$ was not observed with pure Ni sample obtained from ultrasonic spray method (Fig. 4d), regardless of oxidation or reduction conditions. Therefore, without the $Ni-Y_2O_3$ interface, Ni alone could not directly react with $H_2O$ to formed $NiO_x$. The in situ Raman and XRD experiments suggested that when exposed to $H_2O$ vapor, active oxygen species were formed over the $Ni-Y_2O_3$ interface and subsequently converted into $Ni-NiO-Y_2O_3$ interface as the active states for WGS reaction. Therefore, the roles of $Y_2O_3$ were: (I) restricting the growth of the Ni particles under the reduction conditions; (II) forming the $Ni-NiO_x-Y_2O_3$ interfacial sites under the WGS condition.

**Reaction pathway with $Ni-NiO_x-Y_2O_3$ interface participation.** At present, there were mainly two recognized mechanisms for the WGS reaction: redox mechanism and associative mechanism[51–54]. For the redox mechanism, CO reacted with the surface oxygen to form $CO_2$ and vacancies, where $H_2O$ dissociated to form $H_2$. Following this mechanism, the produced $H_2$ could be detected during the $H_2O$ dissociated process. For the associative mechanism, CO and $H_2O$ adsorbed on the surface of the catalyst to form intermediate, and then the intermediate further decomposed to produce $CO_2$ and $H_2$ simultaneously. In that case, $H_2O$ tended to form surface hydroxyl species, which could react with CO[20,21]. Here, the $H_2O$ dissociation experiment was conducted (Supplementary Fig. 11). No $H_2$ was produced in either the thermostatic test or the temperature-programmed heating test, ruling out the existence of redox mechanism. Furthermore, the temperature programmed surface reaction (TPSR) experiment was performed. The results showed a wide range of desorption signals of $CO_2$ (Supplementary Fig. 12), which was caused by the oxidation of CO with surface O. The strong adsorption of CO was also found in the temperature-programmed desorption (TPD) using Ar as the carrier gas (Supplementary Fig. 13) and CO kinetic order (Supplementary Fig. 14c).

From the results of transient isothermal surface reaction (250 °C, Fig. 5a) after WGS reaction, it could be observed that the concentration ratio of $CO_2/H_2$ was ~2 when the reaction reached a stable state. It suggested that $H_2O$ might be dissociated to OH* first, and then OH* reacted with CO to generate $CO_2$ and H*, following the associative mechanism[20,21] according to 2 OH + 2 CO → $H_2$ + 2 $CO_2$. In addition, the cyclic CO temperature-programmed reduction (CO-TPR) was used to verify the production of hydroxyl[36]. As shown in Fig. 5b, the catalyst was treated in the WGS reaction atmosphere at room temperature between two CO-TPR experiments. For the CO-TPR-1, there were mainly three reduction peaks, among which the one at lower temperature (182 °C) was the reduction of small-size NiO, and the one at higher temperature (413 °C) should be the reduction of large-size NiO. The reduction peak at 253 °C was accompanied by the generation of $H_2$, which resulted from the reaction between CO and hydroxyl species[20,21,36]. The formation of hydroxyl was caused by the dissociation of $H_2O$ in the WGS reaction before CO-TPR-1. For CO-TPR-2, two main reduction peaks could be observed, among which the higher temperature reduction peak was considered as the reduction of $NiO_x$ species formed in the reaction atmosphere. This reduction was similar to the $H_2$-TPR result of the used sample (Supplementary Fig. 15a). Accordingly, the reduction peak at 252 °C was also accompanied by the generation of $H_2$ signal, indicating the generation of surface hydroxyl was recyclable. The same result was also observed for the catalyst treated in the $H_2O$/Ar atmosphere between CO-TPR-1 and CO-TPR-2 (Supplementary Fig. 15b). In order to ensure that pure $Y_2O_3$ did not contribute to the dissociation of $H_2O$, we

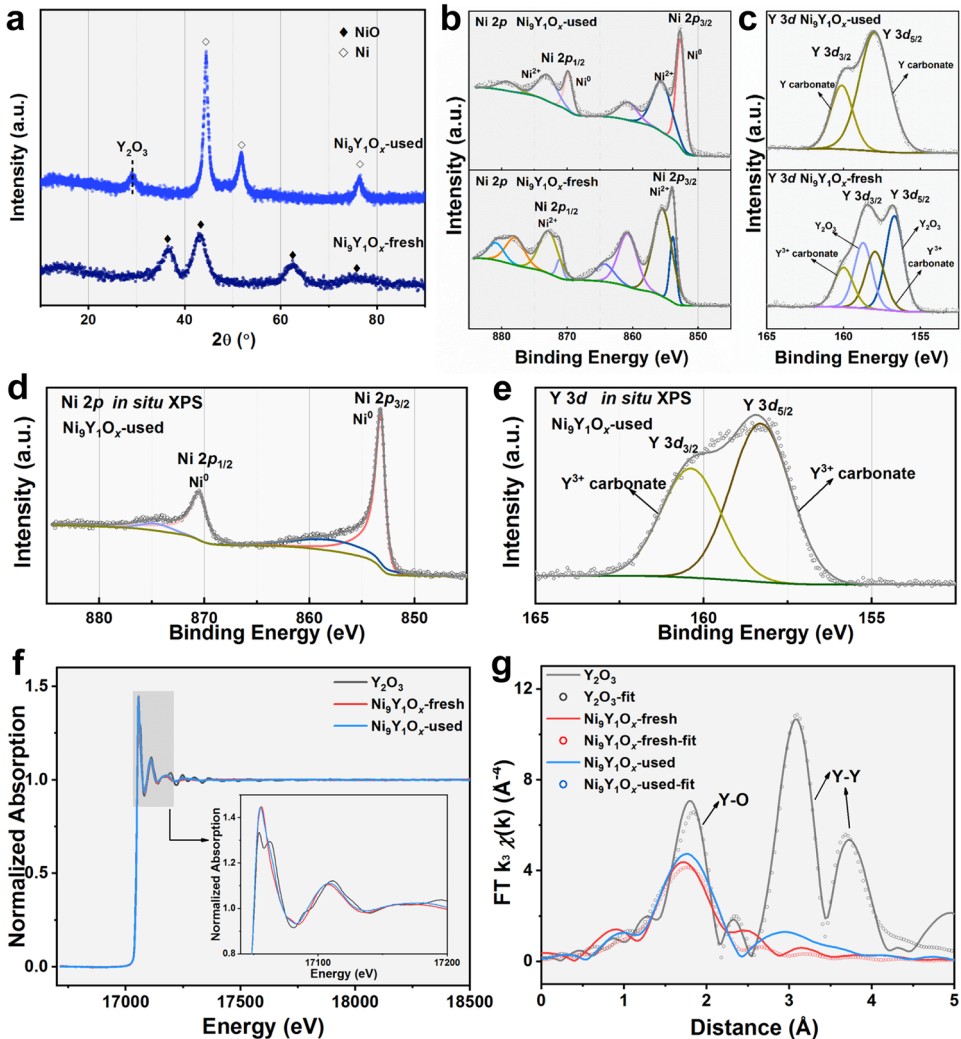

**Fig. 3 Phase change and surface electronic structure. a** XRD patterns of the fresh and used $Ni_9Y_1O_x$ catalysts; **b**, **c** XPS results of the fresh and used $Ni_9Y_1O_x$ catalysts; **d**, **e** quasi in situ XPS results of $Ni_9Y_1O_x$-used; **f** Y $K$ edge X-ray absorption near edge spectra profiles and **g** Y $K$ edge EXAFS of the fresh and used $Ni_9Y_1O_x$ catalysts for experiment data (solid) and fitted lines (dotted).

demonstrated this by activity test, in situ Raman and $H_2O$ dissociation experiments of pure $Y_2O_3$ (Supplementary Fig. 16). It was believed that the interface played an important role in the dissociation of $H_2O$.

We further calculated the activation of the $H_2O$ (300 °C) at different surface locations to verify the hypothesis of interfacial active sites. $H_2O$ adsorbed weakly on metallic Ni surface (Ni{111}). When the partial pressure of $H_2O$ was equal to 10 kPa, the $\Delta G$ of adsorbing on the top position and the bridge position were 0.516 eV and 0.758 eV, respectively. (Supplementary Fig. 17). However, the adsorption of $H_2O$ molecules in the $Y_3O_4/NiO_x/Ni\{111\}$ model was completely different. The $\Delta G$ of $H_2O$ adsorption at the three Y sites were preponderant (−0.428, −0.468 and 0.219 eV) because $Y_2O_3$ was a stronger Lewis acid than Ni (Fig. 5c). The coordination numbers of the Y atom at these three sites were 5, 4, and 4 in turn, and the Bader charges value were 1.84, 2.01 and 2.01, respectively. This was agreement with the EXAFS result of the used sample (the C.N. for Y-O was 5.155 ± 0.392). This result indicated that the lower coordination number (the C.N. of Y atom in $Y_2O_3$ was 6) and higher positive charge of Y atoms led to its stable binding with $H_2O$ molecules. Besides, two models (the Ni{111} and $Y_3O_4/NiO_x/Ni\{111\}$) for $H_2O$ dissociation were also different. For the Ni{111}, although the adsorbed $H_2O$ molecules dissociated to *OH and *H were the

process of decreased $\Delta G$, the whole process of $H_2O$ adsorption and dissociation was still a non-spontaneous process ($\Delta G > 0$) (Supplementary Fig. 17). For the $Y_3O_4/NiO_x/Ni\{111\}$ model, although the $H_2O$ adsorption at site I (−0.428 eV), site II (−0.468 eV) and site III (0.219 eV) were various, the Gibbs free energies of the decomposition products (IMI-2, IMII-2 and IMIII-2) were significantly lower than those of the $Y_3O_4/NiO_x/Ni\{111\}$ plus vapor, i.e., the $\Delta G$ were down around 0.6 eV (Fig. 5c). In general, the $Ni\text{-}NiO_x\text{-}Y_2O_3$ interface was much more favorable for the activation of $H_2O$ than pure Ni. The results of $H_2O$ reaction order measurements also supported the same conclusion (Supplementary Fig. 14a, b). The $H_2O$ reaction order of the pure Ni sample (0.57 at 270 °C) was obviously larger than that of the $Ni_9Y_1O_x$ sample (0.32 at 270 °C), indicating that the latter was easier to adsorb and dissociate $H_2O$. The whole reaction path of the WGS reaction was simulated and presented in Fig. 5d and Supplementary Table 7. $H_2O$ was adsorbed on the atom Y at the first step and CO was adsorbed on the Ni atom at third step. The rate-determining step was the fourth elementary reaction, i.e., *OH migration from atom Y to atom C, and the most probable cause was the bonding force of CO were too strong. As a whole, the WGS reaction was spontaneous because the sign of $\Delta G$ was negative (−1.476 eV at 300 °C, the partial pressure of each gas species was set to the value at the outlet of reactor, shown as Supplementary Table 6). In addition, the $\Delta G$ at the inlet of reactor

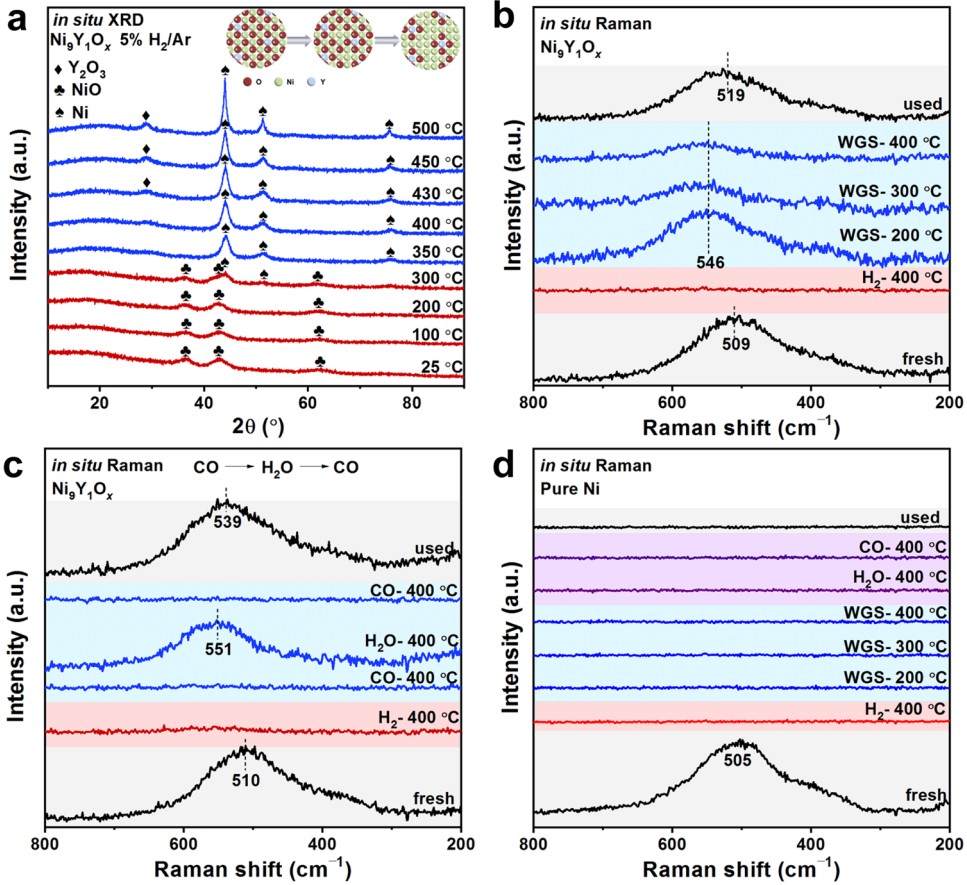

**Fig. 4 In situ characterizations to monitor the interfacial structure changing. a** In situ XRD patterns in 5% $H_2$/Ar for the $Ni_9Y_1O_x$ catalyst; in situ Raman spectra under the WGS reaction conditions for (**b**, **c**) $Ni_9Y_1O_x$ catalyst and (**d**) pure Ni sample.

were also provided and the larger spontaneous tendency was presented for the lower product concentrations (shown as Supplementary Fig. 18 and Supplementary Table 8).

From the conclusion of the reaction pathway study, we proposed the associative reaction mechanism involving the $NiO_x$ species at the interface (Supplementary Fig. 19). The adsorbed $H_2O$ molecules first dissociated at the interface $NiO_x$ site to generate two OH* molecules, and then the active hydroxyl species further reacted with the adsorbed CO to generate $CO_2$ and $H_2$. Therefore, the reactive Ni-$NiO_x$-$Y_2O_3$ interfacial site enabled the catalyst to efficiently catalyze the WGS reaction and largely improved its reaction efficiency.

## Discussion

In summary, the addition of $Y_2O_3$ was essential for the active Ni-$NiO_x$-$Y_2O_3$ interface in WGS reaction. Compared to pure Ni sample, the presence of $Y_2O_3$ tremendously improved the catalytic activity and stability, enabling efficient WGS reaction at medium temperature range. Based on experimental investigation and DFT calculation, the Ni-$NiO_x$-$Y_2O_3$ interface effectively dissociated $H_2O$ molecules and facilitated the reaction with associative mechanism. $Y_2O_3$ and other rare earth metal oxide system should play an important role in catalytic reactions, in particular those with Lewis base molecules such as $H_2O$ and $NH_3$. This will be a golden era for the adventure of rare earth catalysis.

## Methods

**Synthesis of catalysts**. In a typical synthesis of $Ni_aY_bO_x$, 4 mmol nitrates (99%, Tianjin Kermal Factory) were added to 60 mL anhydrous ethanol. The added mass of $Ni(NO_3)_2$·6$H_2O$ and $Y(NO_3)_3$·6$H_2O$ was according to the molar ratio of Ni to Y (1: 9, 3: 7, 5: 5, 7: 3, 9: 1 and 9.5: 0.5). The mixture was stirred at room temperature

for 10 minutes and then dispersed by ultrasound for 10 minutes. Then, $N_2$ flow was used to take small droplets atomized by the atomizer into the high-temperature tubular furnace (450 °C) for decomposition reaction, and the samples were pumped to the receiver for collection. The collected samples were dried overnight in an oven at 60 °C, and then calcined at 400 °C (2 °C min$^{-1}$ of ramping rate). The obtained sample was named $Ni_aY_bO_x$, where a: b was the molar ratio of Ni and Y. Other composite catalysts ($Ni_9Ce_1O_x$, $Ni_9Sm_1O_x$ and $Ni_9Al_1O_x$) and oxides (NiO and $Y_2O_3$) followed the same synthetic method with $Ni_aY_bO_x$, where only nitrates were altered.

**$N_2$ adsorption-desorption measurements**. The $N_2$ adsorption-desorption measurements was proceeded on a Builder SSA-4200 surface area analyzer at 77 K after degased the $Ni_aY_bO_x$ catalysts at 200 °C for 6 h under vacuum. The specific surface area ($S_{BET}$) was calculated from Brunauer-Emmett-Teller (BET) method.

**Transmission electron microscopy (TEM)**. The TEM images were taken on a JEOL JEM-2100F microscope operating at 100 kV. The High-angle annular dark-field scanning transmission electron microscopy (HAADF-STEM) images were obtained on a JEOL ARM200F microscope equipped with a probe-forming spherical-aberration corrector and Gatan image filter (Quantum 965). The element mapping results were derived from the electronic energy-loss spectroscopy (EELS) analysis equipped with STEM model.

**Scanning electron microscopy (SEM)**. The SEM images were taken on a Zeiss SUPRA 55 scanning microscope with the acceleration voltage of 5.0 kV.

**X-ray diffraction (XRD)**. Both ex situ and in situ modes were carried out on a PANalytical X'pert3 powder diffractometer (40 kV, 40 mA) using Cu Kα radiation ($\lambda$ = 0.15406 nm). The powder samples were placed inside a quartz holder and the diffraction angles (2$\theta$) ranged from 10 to 90°. For the in situ XRD experiments, 100 mg catalysts were loaded into an Anton Paar XRK900 in situ chamber. The XRD data was measured from 25 to 450 °C in 5% $H_2$/Ar mixture (30 mL min$^{-1}$) or 2% CO/Ar mixture (30 mL min$^{-1}$).

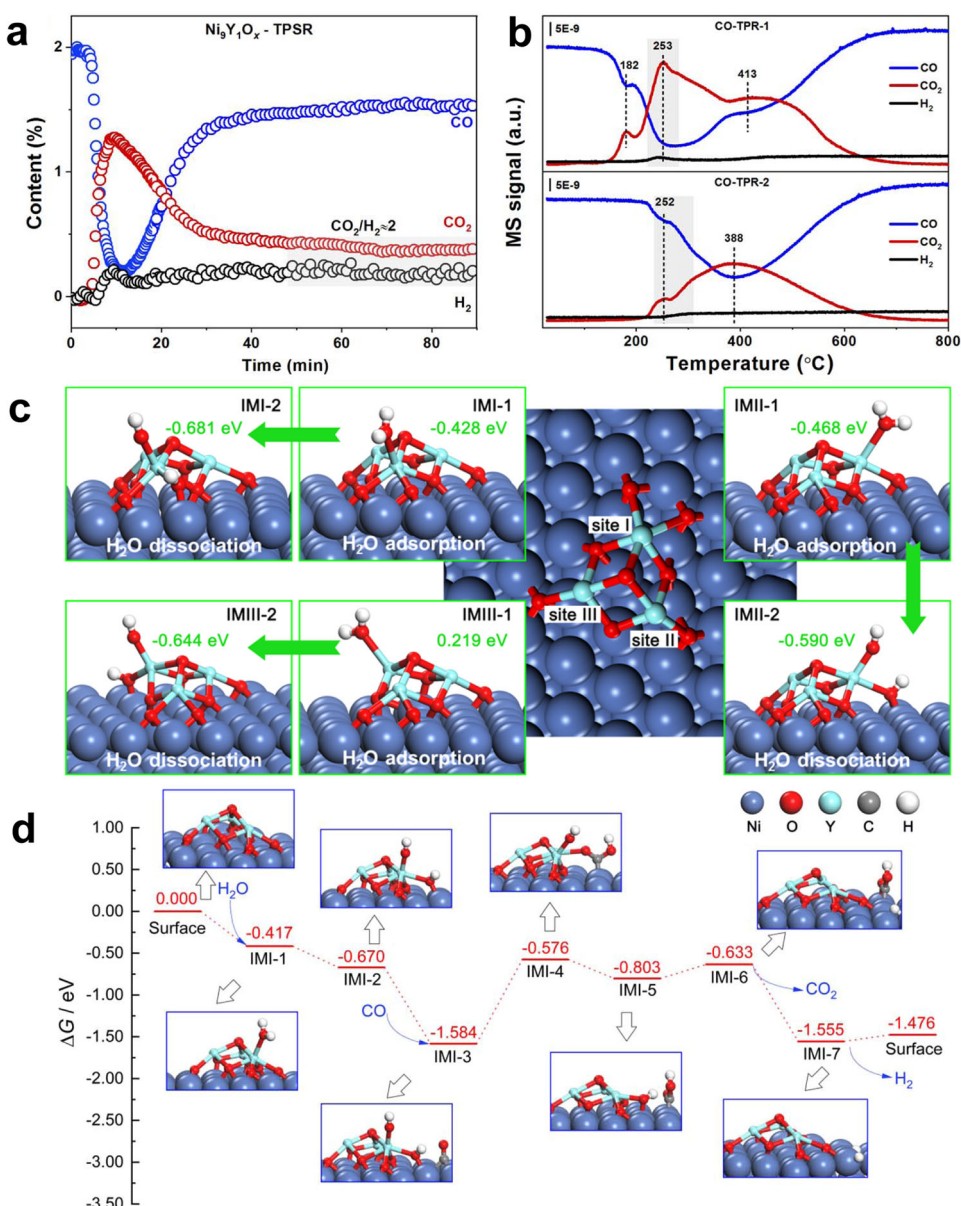

**Fig. 5 The WGS mechanism study and DFT calculation of the Ni₉Y₁Oₓ catalyst. a** Surface reaction on the $Ni_9Y_1O_x$ catalyst at 250 °C under 2%CO/Ar atmosphere; **b** cyclic CO-TPR experiments for CO consumed and $CO_2$ evolved against temperature for the $Ni_9Y_1O_x$ catalyst (Between CO-TPR-1 and CO-TPR-2, the catalyst was treated in the WGS atmosphere at room temperature); **c** the $\Delta G$ of $H_2O$ adsorption and dissociation process at three Y atom sites in the $Y_3O_4/NiO_x/Ni\{111\}$ model, the reaction temperature was set to 300 °C and the partial pressure of water vapor was set to 10 kPa, i.e., the inlet pressure; **d** the simulated WGS reaction pathways (the reaction temperature: 300 °C, the partial pressure of CO, $H_2O$, $CO_2$ and $H_2$: 1, 8000, 2000 and 2000 Pa, i.e., the outlet pressure of the reactor).

**X-ray photoelectron spectroscopy (XPS)**. XPS characterization was carried out on a Thermo scientific ESCALAB Xi⁺ XPS spectrometer with Al Kα radiations, and with the C 1 s peak at 284.8 eV as an internal standard for all the spectra. For the quasi in situ XPS experiments, 20 mg catalysts were loaded into an HPGC 300 reaction chamber made by Fermion instruments (Shanghai) Co., LTD. The XPS data was measured after the in situ WGS reaction (400 °C, 2%CO/~3%H₂O in Ar, 15 mL min⁻¹).

**X-ray absorption fine structure (XAFS)**. Y K-edge (17.038 keV) XAFS was performed at the BL01B1 beamline of Super Photon ring − 8 GeV (SPring-8, Japan). XAFS spectra (16.714–18.499 keV) were measured in transmission mode using Si (311) DCM and ion chambers. Y₂O₃ was used for energy shift calibration. XAFS data was analyzed using Demeter software package (including Athena and Artemis, version 0.9.26). Athena was used for data normalization and Artemis was used for EXAFS fitting.

**Temperature-programmed reduction**. The temperature-programmed reduction by H₂ (H₂-TPR) was carried on a Builder PCSA-1000 instrument. 30 mg catalysts (20 − 40 mesh) were pretreated in air at 300 °C for 0.5 h and then purged with Ar. The final test was carried out in 5%H₂/Ar (30 mL min⁻¹) from room temperature to 700 °C. The temperature-programmed reduction by CO (CO-TPR) was carried on an online mass spectrometer (TILON LC-D200M). 100 mg catalysts (20 − 40 mesh) were pretreated by 5%H₂/Ar (30 mL min⁻¹) at 400 °C for 0.5 h and then purged with Ar. After the WGS reaction at 300 °C for 2 h, the catalyst was heated from room temperature to 800 °C in the 2%CO/Ar (30 mL min⁻¹) to obtain the result of CO-TPR-1. Next, the sample were cooled to room temperature and switched to the retreatment gas (2%CO/~3%H₂O in Ar at RT, 30 mL min⁻¹; or ~3%H₂O in Ar at 300 °C, 30 mL min⁻¹) for 0.5 h. Then, the catalyst was heated from room temperature to 800 °C to obtain the result of CO-TPR-2.

**In situ Raman spectroscopy**. All the Raman spectra were collected from a Lab-RAM HR800 spectrometer (HORIBA JY) with 632.8 nm laser. Samples were placed

on the micro-Raman reaction cell (Xiamen TOPS) and Raman shift ranged from 200 to 800 cm$^{-1}$ with a spectral resolution of 2 cm$^{-1}$. For the first mode, the samples were pretreated by 5%H$_2$/Ar (30 mL min$^{-1}$) at 400 °C for 0.5 h, and then cooled to room temperature. Next, WGS reaction gas (2%CO/~3%H$_2$O in Ar, 30 mL min$^{-1}$) was fed and the samples were heated from room temperature to 400 °C. The second mode was the switch experiment. After pretreated in 5%H$_2$/Ar (30 mL min$^{-1}$) at 400 °C, the samples were purged with two kinds of feeds (2%CO/Ar, 30 mL min$^{-1}$; or ~3%H$_2$O in Ar, 30 mL min$^{-1}$) at 400 °C in turn.

**Temperature-programmed surface reaction (TPSR).** The TPSR experiment was measured by the self-constructed device, using an online mass spectrometer (TILON LC-D200M) as the detector. The experiments were mainly divided into two parts. In the first part, the H$_2$O dissociation experiment, 100 mg catalysts (20 − 40 mesh) were pretreated by 5%H$_2$/Ar (30 mL min$^{-1}$) at 400 °C for 0.5 h and then purged with Ar. After WGS reaction at 300 °C for 2 h, the samples were heated from room temperature to 600 °C under the ~3%H$_2$O/Ar atmosphere to get the heating process H$_2$O dissociation result. The constant temperature H$_2$O dissociation experiment process was carried out after the WGS reaction. ~3%H$_2$O/Ar was fed at a constant temperature (300, 350 and 400 °C) to collect the results. The second part was the reaction of CO with surface OH. After the same pretreatment and in situ reaction process, Ar purge was carried out at 300 °C for 2 h. Then, the samples were cooled to room temperature and switched to the 2%CO/Ar. Next, the results were collected at constant temperature (250 °C).

**The density functional theory (DFT) calculations.** The heterogeneous catalysis simulations were carried out using the Vienna ab Initio simulation package[55–58]. The Perdew−Burke−Ernzerhof (PBE) exchange−correlation functional and the projector augmented-wave (PAW) pseudopotential were applied to spin-unrestricted geometry optimizations[58–60]. The cutoff energy for the plane wave basis was set to 400 eV. The convergence threshold of the electronic self-consistency was specified as 1.0×10$^{-6}$ eV, and the total energy change of the whole catalyst system between two ionic relaxation steps was designated as less than 0.02 eV. The cleaved Ni {111} surface comprised a 5-layer slab (the bottom 2 layers were fixed) as well as a vacuum layer of 15 Å. The lattice parameters of the Ni {111} surface were a = b = 12.4592 Å, c = 23.1383 Å, α = β = 90° and γ = 120°, and the number of Ni atoms was 125. Four oxygen atoms were positioned on the top layer of Ni to simulate NiO$_x$, then Y$_3$O$_4$ cluster was loaded on NiO$_x$ layer, shown as Supplementary Fig. 8. As a group, the formula of the Y$_3$O$_4$/NiO$_x$/Ni{111} model was Ni$_{125}$O$_8$Y$_3$. The binding force between Y$_3$O$_4$ cluster and NiO$_x$ was Y-O or Ni-O ionic bonds, and the NiO$_x$ layer and Ni{111} was bound by metallic bond.

**Catalytic performance measurement.** The WGS activities of the Ni$_a$Y$_b$O$_x$ catalysts were tested in a self-constructed fixed-bed flow reactor. The temperature controller (UDIAN, XIAMEN YUDIAN AUTOMATION TECHNOLOGY CO., LTD.) was used in the reactor temperature control system. Both the actual temperature of the furnace temperature and the location of the catalyst were detected. The gas path system consisted of two parts: CO was the standard gas, and H$_2$O was generated by vaporization after quantitative pumping by water injection pumps. The actual reaction gas content was 2%CO, 10%H$_2$O, and the equilibrium gas was N$_2$. During the test, 100 mg catalysts (20 − 40 mesh) were filled with gas hourly space velocity (GHSV) was 42,000 cm$^3$ g$_{cat}$$^{-1}$ h$^{-1}$. The catalysts were pretreated in a 5% H$_2$/N$_2$ at 400 °C. After a cooling system, the dry gas entered the gas analyzer (Gasboard-3100, Wuhan Sifang Corp), and then the real-time CO, CO$_2$ and CH$_4$ contents were obtained. The CO conversion and CH$_4$ selectivity were calculated through Eq. (1) and Eq. (2), respectively.

$$X_{CO} = \frac{n_{CO}^{in} - n_{CO}^{out}}{n_{CO}^{in}} \times 100\% \tag{1}$$

$$S_{CH_4} = \frac{n_{CH_4}^{out}}{n_{CO_2}^{out} + n_{CH_4}^{out}} \times 100\% \tag{2}$$

$$TOF = \frac{CO\ converted\ (mol)}{Ni\ atom\ at\ interface\ (mol) \times time\ (s)} \tag{3}$$

The stability test of the Ni$_9$Y$_1$O$_x$ catalyst was conducted at 300 °C (GHSV = 420,000 cm$^3$ g$_{cat}$$^{-1}$ h$^{-1}$) for 100 h. The reaction rate was measured using 5 mg catalysts mixed with 200 mg quartz sand, and the CO conversion was controlled below 15%. The turnover frequency (TOF) value were calculated according to the Eq. (3). The kinetic order of CO and H$_2$O was measured by adjusting the content of CO and H$_2$O in the mixed gas at constant temperature (240−280 °C).

## Data availability

The main data supporting the findings of this study are available within the article and its Supplementary Information. All other relevant source data are available from the corresponding author upon reasonable request.

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

## Acknowledgements

This work was financially supported from the National Science Foundation of China (no. 21771117, 21805167, 22075166), the Taishan Scholar Project of Shandong Province of China, the Young Scholars Program of Shandong University (grant nos. 11190089964158), EPSRC (EP/P02467X/1 and EP/S018204/2), Royal Society (RG160661, IES\R3\170097, IES\R1\191035, IEC\R3\193038). We acknowledge SPring-8 (Japan) for the XAFS experiments conducted under the proposal no. 2021A1387 and Dr. Hiroyuki Asakura from Kyoto University for helping with the XAFS measurement. We thank the Center of Structural Characterizations and Property Measurements at Shandong University for the help on sample characterizations.

## Author contributions

C.-J.J. and F.R.W. supervised the work; K.X., H.Y., F.R.W. and C.-J.J. designed the experiments, analyzed the results and wrote the manuscript; K.X. and W.-W.W. performed the in situ XRD, in situ Raman and quasi in situ XPS; S.-Q.L. performed the DFT calculation; K.X., Q.-L.M., W.-P.S. and G.-H.D. performed the catalysts preparation, catalytic tests and the TPR tests; C.M. performed the aberration-corrected HAADF-STEM measurements and analyzed the results. H.G. and F.R.W. performed the XAFS experiments and analyzed the data.

## Competing interests

The authors declare no competing interests.
