## [Peer Review File · Nature Communications]

Title: Catalytically Efficient Ni-NiOx-Y2O3 Interface for Medium Temperature Water-Gas Shift ReactionREVIEWER COMMENTS

Reviewer #1 (Remarks to the Author):

This is an interesting and well-written paper dealing with the catalytic behaviour of the Ni-NiOx-Y2O3 system in the medium temperature water-gas shift reaction. However, in my opinion it does not reach the level to deserve publication in Nature Communications. One of the main reasons is that the catalytic study is carried out under an idealistic gas composition stream containing only CO and H2O. What would happen with a realistic reformat stream containing also CO2 and H2? This study is needed to assess the actual performance of the studied systems.

Some other issues that should be taken into account to improve the paper are following:

1. Line 94. "...and the Ni9Y1Ox with different Ni/Y ratios...". The Ni/Y ratio is always 9/1 in this catalyst. Is this a mistake? Actually, Supplementary Fig. 2 shows the catalytic behaviour of samples Ni_aY_bO_x.
2. Line 169. "During the reduction of H2," Has to be replaced with "During the reduction with H2, ..."
3. Line 289. "Discussion" section should be better the "Conclusions" section.

Reviewer #2 (Remarks to the Author):

Title: Catalytically Efficient Ni-NiOx-Y2O3 Interface for Medium Temperature Water-Gas Shift Reaction. In this manuscript, the authors reported a formation of Ni-NiOx-Y2O3 interface for efficient WGS activity. The formation and presence of NiOx phase in partially reduced Ni-Y2O3 catalysts is confirmed using various in-situ and ex-situ techniques. The manuscript is well organized and explored to highlight the role of the presence of NiOx phase in WGS reaction. The authors observed few interesting results, however, there is a need to understand the actual structure and role of both NiOx and Y2O3 during WGS reaction. For this, I suggest following comments before it can be accepted for the publication in the journal.

1. For activity comparison with the literature (Fig 1), it is required to compare in terms of TOF. The rate of CO converted is highly depends on the amount of available metal centres. Rate per cite shows the activity of the cite.
2. In H2-TPR experiments (Fig S3, S4), the reactant consumption is mainly assigned to conversion of Ni centers, but it is also possible for the consumption by Y2O3 centers. In order to quantify the consumption due to Ni centers and Y2O3 centers, it is suggested to present the Ni-based reducibility for the catalysts. The consumption higher than the theoretical value for the available Ni species should be from Y2O3 species.
3. XAFS results, when there is a Ni-Y2O3 interphase formation, it is expected to have Ni-O-Y or Ni-Y bonds. Identifying these bonds supports the interphase formation in reduced and used catalysts.
4. In Fig 4, Raman study, it is required to perform on Y2O3 support as well, the role of Y2O3 in activating H2O during WGS reaction should be compared with Ni-Y2O3 catalyst. Furthermore, in this section both

XRD and Raman are bulk techniques the position of NiO_x in Ni-Y₂O₃ is hard to confirm. It is suggested to consider surface techniques such as identifying the lattice from TEM images should support the above observations.

5. To understand the reaction pathway for WGS reactions, in situ DRIFTS experiments helps to identify the intermediates formed at WGS reaction condition. If possible the authors should consider CO-DRIFTS or WGS-DRIFTS experiments for the selected samples.

Reviewer #3 (Remarks to the Author):

An interesting paper in an important topic. There are several key points that need justification or a better explanation.

1) On page 3, the authors should mention the performance of other WGS catalysts that do not contain Ni or a metal/oxide combination. A Au/MoC catalyst has been reported that can accelerate the WGS at temperatures much lower than 300 C (Yao et al, Science, 357 (2017) 389).

2) On page 10, what exactly is the nature of the interaction between Ni/NiO and Y₂O₃?

3) On pages 13 and 14, the model used in the DFT calculations needs a much better justification. What is the exact link between the model and the catalyst samples used to do the experiments?

4) Top of page 15, OK the addition of Y₂O₃ enhances the catalytic performance of Ni for the WGS reaction. Why that finding should be published in top-level journal like Nature Communications? It looks like a nice finding for a regular journal in catalysis.

Responses to the Reviewers' Comments and the Corresponding Revisions

To Reviewer 1:

Reviewer #1: This is an interesting and well-written paper dealing with the catalytic behaviour of the Ni-NiO_x-Y₂O₃ system in the medium temperature water-gas shift reaction. However, in my opinion it does not reach the level to deserve publication in Nature Communications. One of the main reasons is that the catalytic study is carried out under an idealistic gas composition stream containing only CO and H₂O. What would happen with a realistic reformat stream containing also CO₂ and H₂? This study is needed to assess the actual performance of the studied systems.

Response: Thanks for the reviewer's valuable comments and suggestions. We have performed additional catalytic study with the presence of CO₂ and H₂, as shown in Table R2. This manuscript provides new concept of interface design, and put a range of early transition fifth and sixth row metal oxides on to the screening radar for catalysis research. The ability to activate thermodynamically stable molecules such as H₂O and CO₂ is the key innovation of this research.

First of all, the construction of catalytically active interfaces between metals and oxides has attracted a lot of attention. The construction and application of metal-rare earth oxides (except for CeO₂) interfaces are not yet mature. These early transition rare earth cations are strong Lewis acid, which facilitates its adsorption of molecules, such as H₂O. **Exploring the role of such interfaces is important for environmental catalysis, energy catalysis, and the applications of rare earth materials.** In addition, H₂O dissociation usually considered as the rate-determining step for the water gas shift (WGS) reaction, which is beneficial for us to investigate the Lewis base molecular activation of the interface. For Ni-based catalysts, decreasing application temperature of Ni-based catalysts to the medium-temperature (MT) region can achieve rapid kinetics with high CO conversion equilibrium. **Therefore, it is significant to prepare a composite rare earth Ni-based catalyst for MT-WGS reaction which contains an efficient catalytic interface.** Moreover, the dynamic structure change and catalytic effect of such interface during the reaction is needed to clear understanding.

In this manuscript, we have successfully constructed sufficient and stable Ni-NiO_x-Y₂O₃ interfacial sites, which exhibit excellent catalytic performance for MT-WGS reaction. **We believe our findings will interest a broad readership in catalysis and materials science according to the following aspects:**

1. We have successfully prepared Ni-Y₂O₃ composite catalyst and transferred the WGS reaction operating temperature from HT to MT for Ni-based catalyst. The reaction rate of the Ni-Y₂O₃ catalyst with Ni-NiO_x-Y₂O₃ interfacial structure reached 140.6 μmol_{CO} g_{cat}⁻¹ s⁻¹ at 300 °C, achieving the highest activity of Ni-based catalysts. Besides, it exhibited solid stability under harsh reaction conditions (300 °C, space velocity of 420,000 cm³ g_{cat}⁻¹ h⁻¹).

2. We created the catalytically efficient Ni-NiO_x-Y₂O₃ interface sites in the Ni-Y₂O₃ catalyst and it was preserved during the WGS reaction process. We clearly observed the formation of Ni-Y₂O₃ interface, and the dynamic structure change of the Ni-NiO_x-Y₂O₃ interface was clearly recorded by *in situ* Raman. These findings indicated that the real active interface structure in the WGS reaction was precisely proved.

3. We demonstrated the realistic role of the Ni-NiO_x-Y₂O₃ interfaces during the catalytic process. Based on the experimental investigation and DFT calculation, the Ni-NiO_x-Y₂O₃ interfaces effectively dissociated H₂O molecules and facilitated the reaction in accordance with the associative mechanism. Thus, the significance of the Ni-NiO_x-Y₂O₃ interface for catalytic performance, as well as the reaction mechanism was adequately proved.

After all, this manuscript focuses on the construction, dynamic change and catalytic effect of the Ni-NiO_x-Y₂O₃ interface, which represents an important findings in the fields of catalysis, materials and surface research. In response to the suggestion of reviewer that the catalytic performance test of Ni₉Y₁O_x catalyst in realistic stream, we have also conducted adequate literature comparison and supplementary experiments.

Extensive previous work about the WGS reactions have been mainly conducted using CO and H₂O as the reaction atmosphere to evaluate catalytic performance¹⁻⁵, which also includes Ni-based catalysts⁶⁻¹⁰. The main factors of Ni-based catalysts causing excellent

WGS performance have been explained in reports. Ang, M. L. *et al.* achieved improved catalyst activity and H₂ yield by doping the Ni/CeO₂ catalyst with alkali metals, such as Na and K (Ang, M. L. *et al.* *ACS Catal.* **4**, 3237–3248 (2014); Ang, M. L. *et al.* *J. Catal.* **329**, 130–143 (2015).). Saw, E. T. *et al.* constructed a Cu-Ni/CeO₂ bimetallic catalyst and found that the formation of Ni-Cu alloy prevented the dissociation of CO, which in turn prevented the formation of carbon species “formate”, thus inhibiting the formation of CH₄ as an undesired side product (Saw, E. T. *et al.* *J. Catal.* **314**, 32–46 (2014).). Xu, M. *et al.* prepared Ni nanoparticle catalysts modified by TiO_{2-x}, which were found to exhibit excellent catalytic activity and low CH₄ selectivity at 350 °C. The presence of Ni^{δ-}-O_v-Ti³⁺ active site was identified by XPS and XAFS, which was considered to be responsible for the excellent catalytic performance of the catalyst (Xu, M. *et al.* *ACS Catal.* **7**, 7600–7609 (2017); Xu, M. *et al.* *J. Am. Chem. Soc.* **140**, 11241–11251 (2018).). **Importantly, the descriptions of the catalytic performance (CO conversion, CH₄ selectivity, etc.) in many reports were chosen for atmospheres containing only CO and H₂O (Table R1).** In order to make our study more comprehensive and competitive, **we have added experiments in the realistic WGS reaction stream at 300 °C (containing CO, H₂O, CO₂, H₂) to the study.** The specific experiments are as follows (Table R2):

Table R1. A list of reaction stream in the literature that contain only CO and H₂O.

Sample	Reaction gas content	GHSV (cm ³ g _{cat} ⁻¹ h ⁻¹)	T (°C)	CO Conversion (%)	CH ₄ Selectivity (%)	Reference
Inverse CeO ₂ /Cu	2% CO, 10% H ₂ O, N ₂	42,000	300	over 90.0%	0.0	1
Au_Cluster-CeO ₂	2% CO, 12% H ₂ O, N ₂	42,000	300	over 80.0%	0.0	2
0.5 wt % Au/CeO ₂	2% CO, 8% H ₂ O, Ar	163,488	120	3.5%	0.0	3
Ir/FeO _x -2.40	2% CO, 10% H ₂ O, He	18,000	300	over 90.0%	0.0	4
Pt/Na-TiO ₂	3% CO, 10% H ₂ O, He	2,000	300	over 90.0%	0.0	5
Ni@TiO _{2-x} (450)	6% CO, 24% H ₂ O, Ar	66,000	350	99.3%	5.0%	6, 7

Ni/2Na/CeO ₂	5% CO, 25% H ₂ O, He	68,000	400	97.5%	below 0.5%	8
Ni/5K/CeO ₂	5% CO, 25% H ₂ O, He	68,000	400	97.2%	0.0	9
1Ni9Cu/CeO ₂	5% CO, 25% H ₂ O, He	68,000	400	–	0.0	10

Table R2. CO conversion and CH₄ selectivity of Ni₉Y₁O_x catalysts under various streams.

Experiment type	Reaction gas content	GHSV (cm ³ g _{cat} ⁻¹ h ⁻¹)	T (°C)	CO Conversion (%)	CH ₄ Selectivity (%)
Different H ₂ content	5% CO, 23.3% H ₂ O, 5% H₂ , 5% CO ₂ , N ₂	240,000	300	47.03	1.95
	5% CO, 23.3% H ₂ O, 10% H₂ , 5% CO ₂ , N ₂	240,000	300	51.46	9.09
	5% CO, 23.3% H ₂ O, 20% H₂ , 5% CO ₂ , N ₂	240,000	300	65.77	31.53
Different H ₂ O content	5% CO, 15.6% H₂O , 10% H ₂ , 5% CO ₂ , N ₂	240,000	300	46.30	2.73
	5% CO, 23.3% H₂O , 10% H ₂ , 5% CO ₂ , N ₂	240,000	300	47.03	1.95
	5% CO, 31.1% H₂O , 10% H ₂ , 5% CO ₂ , N ₂	240,000	300	68.31	3.21
Different CO ₂ content	5% CO, 23.3% H ₂ O, 5% H ₂ , 3% CO₂ , N ₂	240,000	300	51.72	3.54
	5% CO, 23.3% H ₂ O, 5% H ₂ , 5% CO₂ , N ₂	240,000	300	47.03	1.95
	5% CO, 23.3% H ₂ O, 5% H ₂ , 10% CO₂ , N ₂	240,000	300	44.62	4.37

(1) Investigation on the effect of H₂ concentration in realistic stream to catalytic performance (Figure R1a).

H₂ was the product of the WGS reaction. According to the chemical equilibrium in the WGS reaction, an increase in H₂ concentration was not favourable for CO conversion. For the test in different H₂ concentrations, it was found that the CO conversion increased significantly with increasing H₂ concentration. The corresponding increase in CH₄ selectivity

suggests that the increased H₂ concentration caused methanation side reactions, which led to the larger CO conversion.

Figure R1. The catalytic performance of Ni₉Y₁O_x catalysts in various streams at 300 °C. (a) Different H₂ content; (b) different H₂O content; (c) different CO₂ content; (d) time-on-stream tests of the Ni₉Y₁O_x and commercial Cu-Zn-Al.

(2) Investigation on the effect of H₂O concentration in realistic stream to catalytic performance (Figure R1b).

For the test in different H₂O concentrations, the CO conversion increased with the increase of H₂O concentration. However, the lowest CH₄ selectivity was observed when the

H₂O concentration was 23.3%. Therefore, on the whole, the H₂O concentration of 23.3% was chosen for the relevant study.

(3) Investigation on the effect of CO₂ concentration in realistic stream to catalytic performance (Figure R1c).

CO₂ was the product of the WGS reaction as well as H₂. For the experiments at different CO₂ concentrations, the CO conversion decreases as the concentration of CO₂ increases, which was in accordance with the chemical equilibrium. Besides, the lowest CH₄ selectivity was observed when the CO₂ concentration was 5%. Therefore, the CO₂ concentration of 5% was chosen for the relevant study.

(4) Stability test and comparison with commercial Cu-Zn-Al catalyst in realistic stream (Figure R1d).

In general, the reaction stream of 5% CO, 23.3% H₂O, 10% H₂, 5% CO₂, N₂ was chosen for the relevant studies. Stability tests of the Ni₉Y₁O_x catalyst were performed in this atmosphere as well as for the commercial catalyst Cu-Zn-Al. It was found that these two catalysts had similar CO conversion in this stream. It also maintained good stability at high temperature and with an ultra-high GHSV, in which the conversion decayed from 47% to 40% in 45 h.

In summary, we have explored the effect of the variation of H₂O, H₂ and CO₂ concentration on the catalytic performance. **The Ni₉Y₁O_x catalyst was still competitive when the reaction stream was 5% CO, 23.3% H₂O, 10% H₂, 5% CO₂, N₂. The catalytic performance in this stream was comparable to that of commercial Cu-Zn-Al catalysts.** Because this manuscript focuses on the structural composition of the Ni-NiO_x-Y₂O₃ interfacial and its catalytic effect, the stream containing only CO and H₂O was selected by us. And we focused on the enhancement of the interfacial effect on the dissociation of H₂O. **The Figure R1d has been added as Supplementary Fig. 2b in the revised supporting information on page S6. And the corresponding description has been shown in page 5, line 28–29 in the revised manuscript and page S6, line 11–14 in the revised supporting information (highlighted in yellow).** We were looking forward to your next comments.

Some other issues that should be taken into account to improve the paper are following:

Comment 1: Line 94. “...and the $Ni_9Y_1O_x$ with different Ni/Y ratios...”. The Ni/Y ratio is always 9/1 in this catalyst. Is this a mistake? Actually, Supplementary Fig. 2 shows the catalytic behaviour of samples $Ni_aY_bO_x$.

Response: We greatly thank for the reviewer. The manuscript has been checked and the writing errors are found and corrected. This position should indeed be $Ni_aY_bO_x$, not just limited to $Ni_9Y_1O_x$ catalyst. **The corresponding description has been shown in page 5, line 9 in the revised manuscript.**

Comment 2: Line 169. “During the reduction of H_2 , ...” Has to be replaced with “During the reduction with H_2 , ...”

Response: We greatly thank for the reviewer. The manuscript has been thoroughly checked. This position has been corrected to “During the reduction with H_2 , ...”. **The corresponding description has been shown in page 9, line 21 in the revised manuscript.**

Comment 3: Line 289. “Discussion” section should be better the “Conclusions” section.

Response: Thanks for the reviewer’s suggestion. We strongly agree with the comment of reviewer. In fact, this section is indeed the “Conclusion” section. However, according to “guide to formatting articles” of *Nature Communications*, **it does not contain the “Conclusion” section. Therefore, this section is generally labeled as “Discussion”.**

To Reviewer 2:

Reviewer #2: Title: Catalytically Efficient Ni-NiO_x-Y₂O₃ Interface for Medium Temperature Water-Gas Shift Reaction.

In this manuscript, the authors reported a formation of Ni-NiO_x-Y₂O₃ interface for efficient WGS activity. The formation and presence of NiO_x phase in partially reduced Ni-Y₂O₃ catalysts is confirmed using various in-situ and ex-situ techniques. The manuscript is well organized and explored to highlight the role of the presence of NiO_x phase in WGS reaction. The authors observed few interesting results, however, there is a need to understand the actual structure and role of both NiO_x and Y₂O₃ during WGS reaction. For this, I suggest following comments before it can be accepted for the publication in the journal.

Response: Thanks for the reviewer's valuable comments and suggestions. The composite Ni₉Y₁O_x catalyst prepared in this manuscript formed a Ni-NiO_x-Y₂O₃ interface during the WGS reaction, which led to a significant improvement for the catalytic performance. The structure and role of this interfacial structure was clearly characterized and explained in the manuscript. The structure of Ni-NiO_x-Y₂O₃ interface was demonstrated by HAADF-STEM and *in situ* Raman to show the existence of interfaces and phase changes. The distribution of Ni species and Y₂O₃ in the catalyst after the reaction could be seen from HAADF-STEM as well as the elemental distribution results (Figure R2a). The Y₂O₃ were mainly distributed in the surroundings of Ni particles. This structure prevented the further growth of Ni particles and provided abundant interfaces between Ni and Y₂O₃. Further by *in situ* Raman test, we found that a vibrational peak of NiO_x appeared during the reaction (Fig. 4b), further indicating the presence of a Ni-NiO_x-Y₂O₃ ternary interface. We were looking forward to your next comments.

Comment 1: For activity comparison with the literature (Fig 1), it is required to compare in terms of TOF. The rate of CO converted is highly depends on the amount of available metal centres. Rate per cite shows the activity of the cite.

Response: Thanks for the reviewer's valuable comments and suggestions. We considered the Ni-NiO_x-Y₂O₃ interface as the active center through experimental investigation and theoretical calculation, so we estimated the TOF of Ni₉Y₁O_x catalyst based on the effective Ni atomic at the interface. The percentage of Ni atoms at the interface was calculated as follows:

Based on the results of elemental distribution, we believed that Y₂O₃ was distributed in the surroundings of Ni particles (Figure R2a). In other words, Y₂O₃ did not completely encapsulated the Ni particles, so we believed that the Ni atoms forming the interface were mainly contained in the circumference of the largest diameter of the Ni particles (Figure R2b, c). We have estimated the effective Ni atomic ratio according to this method.

Figure R2. (a) Electron energy-loss spectroscopy (EELS) elemental mapping results of Ni₉Y₁O_x-used; (b) schematic of the interface; (c) schematic of the Ni atoms forming the interface (red) with the whole Ni particles.

Firstly, based on the *in situ* XRD results, we concluded that the size of the Ni particles was 5.5 nm after H₂ pretreatment of the catalyst at 400 °C (Fig. 4a, Supplementary Table 5). Using 5.5 nm as the diameter, the circumference of the circle was calculated as $\pi d = 17.279$ nm. The diameter of Ni atomic was **0.248 nm**. Thus the effective Ni atomic number ($N_{\text{effective}}$) was about **70**.

$$N_{\text{effective}} = \frac{\pi d_{\text{particle}}}{d_{\text{Ni atom}}}$$

Secondly, we estimated the total number of Ni atoms contained in Ni particles. The volume of a 5.5 nm diameter particle was $4/3\pi r^3 = 87.11$ nm³. The density of metallic Ni was **8.902 g/cm³**. Thus the mass of a 5.5 nm diameter particle was **7.753 E⁻¹⁹ g**. The mass of one Ni atom was **9.747 E⁻²³ g**. Therefore, **one 5.5 nm particle had 7954 Ni atoms**.

$$N_{\text{all}} = \frac{m_{\text{particle}}}{m_{\text{Ni atom}}}$$

Next, we could obtain the proportion of effective Ni atoms (at the interface) as **70/7954 = 0.88%**. The number of effective Ni atoms in 1 g catalyst was **1*67.3%/M_{Ni}*0.88%*10⁶ = 100.9 μmol**.

$$\eta = \frac{N_{\text{effective}}}{N_{\text{all}}}$$

$$n_{\text{effective}} = \frac{1 \text{ g} \times 67.3\% \times \eta \times 10^6}{M_{\text{Ni}}}$$

$$\text{TOF} = \frac{\text{reaction rate}}{n_{\text{effective}}}$$

Finally we calculated the TOF of Ni₉Y₁O_x catalyst at different temperatures based on the reaction rate. In addition, we have added a comparison of the TOF values in the cited literature, as shown in Table R3. Its TOF value (350 °C) is higher than that in the literature, which was consistent with the results of the reaction rate comparison. **The related data has been added in Supplementary Table 1 in the revised supporting information on page**

S22. And the corresponding description has been shown in page 5, line 23–24 in the revised manuscript (highlighted in yellow).

Table R3. Comparison of catalytic performances for WGS reaction over various catalysts.

Catalyst	Reaction conditions	Reaction temperature (°C)	Reaction rate ($\mu\text{mol}_{\text{CO}} \text{g}_{\text{cat}}^{-1} \text{s}^{-1}$)	TOF (s^{-1})	Ref.
$\text{Ni}_9\text{Y}_1\text{O}_x$	2% CO, 10% H_2O , balance N_2	250	37.2	0.37	This work
		300	140.6	1.39	
		350	434.7	4.31	
1Re-10Ni/CeO ₂	5% CO, 10% H_2O , balance He	300	30.0	–	11
$\text{Ni}_{20}/\text{CeLaO}_x$	10% CO, 20% H_2O , balance He	300	35.2	–	12
		350	36.3	–	
$\text{Cu}_4\text{Ni}_{16}/\text{CeLaO}_x$	10% CO, 20% H_2O , balance He	300	31.1	–	
		350	35.9	–	
$\text{Ni}@/\text{TiO}_{2-x}$ (450)	6% CO, 24% H_2O , 70% Ar	350	356.8	3.8	6
Inverse CeO ₂ /Cu	2% CO, 10% H_2O , balance N_2	300	47.5	0.058	1
Cu-Ce(La)O_x	2% CO, 10% H_2O , balance N_2	300	9.0	–	13
$\text{Cu}_{0.3}\text{Fe}_{0.7}\text{O}_x$	2% CO, 10% H_2O , balance N_2	300	12.2	–	14
Ir_1/FeO_x	2% CO, 10% H_2O , balance He	300	43.4	2.31	4
5Pt/0.06Na-TiO ₂	3% CO, 10% H_2O , balance He	250	38.4	1.58	5
2% Au/ α -MoC	10.5% CO, 21% H_2O , 20% N_2 , balance Ar	200	325	–	15
0.2Pt-Mo ₂ N	2% CO, 10% H_2O , balance N_2	300	232	–	16

Comment 2: In H_2 -TPR experiments (Fig S3, S4), the reactant consumption is mainly assigned to conversion of Ni centers, but it is also possible for the consumption by Y_2O_3 centers. In order to quantify the consumption due to Ni centers and Y_2O_3 centers, it is suggested to present the Ni-based reducibility for the catalysts. The consumption higher than the theoretical value for the available Ni species should be from Y_2O_3 species.

Response: Thanks for the reviewer’s valuable comments and suggestions. We quantitatively calculated the theoretical and actual hydrogen consumption of the catalysts, and the results are shown in the Table R4. The results showed that the actual hydrogen consumption of all samples was higher than the theoretical value. And the higher values became smaller as the Y content decreased. These phenomena indicated that the redox property of Y_2O_3 was also increased due to the formation of the interface, which confirmed the interaction existed in the Ni-NiO_x-Y₂O₃ interface. **The related data has been added in Supplementary Table 2 in the revised supporting information on page S23. The corresponding description has been shown in page 9, line 26–28 in the revised manuscript and page S7, line 14–19 in the revised supporting information (highlighted in yellow).**

Table R4. The Ni contents, S_{BET} and the H₂ consumption of the Ni_aY_bO_x catalysts.

Sample	Mole ratio of Ni/Y	Ni content (wt %) ^a	NiO content (wt %) ^b	S_{BET} (m ² g ⁻¹) ^c	H ₂ consumption (μmol g _{cat} ⁻¹)	
					Actual ^d	Theoretic
Ni ₉ Y ₁ O _x	9:1	67.3	85.6	55.5	11974	11413
Ni ₇ Y ₃ O _x	7:3	47.7	60.7	27.8	11319	8093
Ni ₅ Y ₅ O _x	5:5	31.3	39.8	25.2	9384	5307
Ni ₃ Y ₇ O _x	3:7	17.4	22.1	22.2	7410	2947
Ni ₁ Y ₉ O _x	1:9	5.4	6.8	12.9	3522	907

^aThe theoretic percentage of metallic Ni in the total weight.
^bThe theoretic percentage of NiO in the total weight.
^cDetermined with N₂ adsorption.
^dThe actual date was determined with H₂-TPR profile (Supplementary Fig. 3).

Comment 3: XAFS results, when there is a Ni-Y₂O₃ interphase formation, it is expected to have Ni-O-Y or Ni-Y bonds. Identifying these bonds supports the interphase formation in reduced and used catalysts.

Response: Thanks for the reviewer’s valuable comments and suggestions. XAFS generally was used in bulk physical environment (Newville, M. Fundamentals of XAFS. *Reviews in Mineralogy & Geochemistry* **78**, 33–74 (2014)). Therefore, it reflected the bulk information of the catalysts. The interfacial structure was a small percentage of the catalyst. The Ni-O-Y

or Ni-Y coordination has been averaged out with the majority of the NiO and Ni in the fresh and used phase, respectively. Therefore we cannot observe the contribution of Ni-O-Y or Ni-Y coordination in the EXAFS fitting. The results of XRD (Fig. 3a), which was also a bulk characterization technique, showed the fresh catalyst was mainly NiO phase, and the used catalyst was mainly Y₂O₃ and Ni. The change of the interfacial phase could not be accurately detected. In this manuscript we provided a clear description of the interface by *in situ* Raman spectroscopy with high sensitivity. In addition, the change in the coordination situation of the fresh and used catalysts was mainly related to the better crystallization of Y₂O₃ after the catalytic reaction, which was consistent with the XRD results (Fig. 3a). **The corresponding description has been shown in page 9, line 18–20 in the revised manuscript.**

Comment 4: In Fig 4, Raman study, it is required to perform on Y₂O₃ support as well, the role of Y₂O₃ in activating H₂O during WGS reaction should be compared with Ni-Y₂O₃ catalyst. Furthermore, in this section both XRD and Raman are bulk techniques the position of NiO_x in Ni-Y₂O₃ is hard to confirm. It is suggested to consider surface techniques such as identifying the lattice from TEM images should support the above observations.

Response: Thanks for the reviewer's valuable comments and suggestions. In order to address the reviewers' doubts about the ability of Y₂O₃ to dissociate H₂O, we designed relevant experiments. **Activity test, *in situ* Raman and H₂O dissociation experiments were carried out for pure Y₂O₃ samples.** Firstly, from the activity test results, pure Y₂O₃ has no catalytic activity for the WGS reaction (Figure R3a). Secondly, it was mentioned in the comments to add the *in situ* Raman of pure Y₂O₃, but the result did not show any valuable information (Figure R3b). It showed that Y₂O₃ did not have any obvious signal due to poor crystallization, and no significant changes after switched to H₂, WGS reaction stream, H₂O and CO atmosphere. We could not directly observe the dissociation of H₂O. So we further verified it using H₂O dissociation experiments (Figure R3c). There was no H₂ signal was observed in the range of 250–350 °C in ~3H₂O/Ar stream, which indicated that pure Y₂O₃ could not effectively dissociated H₂O. The above results also demonstrated that the formation of the Ni-NiO_x-Y₂O₃ interface was very effective for H₂O dissociation.

Besides, the reviewers suggested verifying the formation of NiO_x by the surface techniques such as TEM. In fact, NiO_x was generated in the original WGS atmosphere and the NiO_x was present only at the interface. Moreover, we believe that the NiO_x formation was amorphous and therefore difficult to observe from TEM. To prove the existence of NiO_x, the *in situ* Raman results have been fully verified. Raman was not a complete bulk technique, different laser led to different detection depths. For example, Zhang *et al.* successfully detected a thin layer of MoO_x on the surface of Pt-Mo₂N catalyst by using a laser of 632.8 nm (Figure R4, Zhang, Z.-S. *et al. J. Am. Chem. Soc.* **142**, 13362–13371 (2020)). Therefore, Raman could be used as a sensitive surface probe to demonstrate the formation of NiO-NiO_x-Y₂O₃ interface during the reaction process. **The Figure R3 has been added as Supplementary Fig. 15 in the revised supporting information on page S20. And the corresponding description has been shown from page 13, line 28 to page 14, line 2 in the revised manuscript and page S19, line 6–17 in the revised supporting information (highlighted in yellow).**

Figure R3. (a) The CO conversion of pure Y₂O₃ for the WGS reaction; (b) *in situ* Raman spectra of pure Y₂O₃ in different streams; (c) the H₂O dissociation experiment of pure Y₂O₃ at different temperatures (250, 300 and 350 °C).

Figure R4. *In situ* characterization and simulated structure model of the 0.2Pt–Mo₂N catalyst with different treatment programs. (a) *In situ* Raman spectra in treatment of switchover of O₂, Ar, and H₂ at RT (21% O₂/N₂, pure Ar, and 5% H₂/Ar). (b) *In situ* Raman spectra in the reduction (5% H₂/Ar → Ar, 300 °C) and WGS reaction treatment process (2% CO/3% H₂O/N₂, 150 °C). (c) *In situ* Raman spectra in the renitridation (10% NH₃/Ar, 500 and 150 °C) and WGS reaction treatment process (2% CO/3% H₂O/N₂, 150 °C). (d) Schematic diagram of structure transformation of the surface oxide layers in the form of MoO₃/γ-Mo₂N and MoO_x/γ-Mo₂N for the 0.2Pt–Mo₂N catalyst. The teal, blue, and red spheres represent the Mo, N, and O atoms, respectively. (Zhang, Z.-S. *et al. J. Am. Chem. Soc.* **142**, 13362–13371 (2020).)

Comment 5: To understand the reaction pathway for WGS reactions, *in situ* DRIFTS experiments helps to identify the intermediates formed at WGS reaction condition. If possible the authors should consider CO-DRIFTS or WGS-DRIFTS experiments for the selected samples.

Response: Thanks for the reviewer’s valuable comments and suggestions. The *in situ* DRIFTS in CO (Figure R5a) and WGS reaction stream (Figure R5b) were supplemented as

requested by the reviewer. **However, due to the dark color of the catalyst (Figure R5d), we could not obtain effective adsorption signals and intermediate information, and could only obtain gas signals of CO or CO₂.** In addition, we also used *in situ* infrared spectroscopy in transmission mode (Figure R5c) and tested the adsorption in 10⁻³ mbar CO condition, and there still no significant adsorption signal was found. Even with KBr blending, no valid information was obtained.

Figure R5. (a) *In situ* DRIFTS experiment in 2% CO/He stream at 30 °C; (b) *in situ* DRIFTS experiment in WGS reaction stream (2% CO/~3% H₂O/He) at 300 °C; (c) *in situ* infrared spectroscopy in transmission mode at -130 °C; (d) the color of Ni₉Y₁O_x catalyst.

To Reviewer 3:

Reviewer #3: *An interesting paper in an important topic. There are several key points that need justification or a better explanation.*

Response: Thanks for the reviewer's valuable comments.

Comment 1: *On page 3, the authors should mention the performance of other WGS catalysts that do not contain Ni or a metal/oxide combination. A Au/MoC catalyst has been reported that can accelerate the WGS at temperatures much lower than 300 °C (Yao et al, Science, 357 (2017) 389).*

Response: Thanks for the reviewer's valuable comments and suggestions. For Ni-based catalysts, current reports show that they are mainly used to catalyze high-temperature (HT) WGS reactions (Xu, M. *et al. J. Am. Chem. Soc.* **140**, 11241–11251 (2018); Ang, M. L. *et al. ACS Catal.* **4**, 3237–3248 (2014); Saw, E. T. *et al. J. Catal.* **314**, 32–46 (2014), and so on.). Decreasing application temperature of Ni-based catalysts to the medium-temperature (MT) region could achieve rapid kinetics with high CO conversion equilibrium. In our manuscript, the Ni₉Y₁O_x catalyst had good catalytic performance for the MT-WGS reaction, which was an important advantage of this catalyst. However, for Cu-based and noble metal (Au, Pt, *et al.*) based catalysts, it was mainly the active component for low-temperature (LT) catalysts. For non-oxide supports catalysts, especially MoC or Mo₂N loaded with noble metals, high CO conversion could be observed even below 150°C. Therefore, the applicable temperature range of these catalysts was quite different from that of the Ni₉Y₁O_x catalysts we obtained. In order to make a comprehensive comparison, we added the relevant data of typical LT-WGS reaction catalysts, such as Au/ α -MoC (Yao, S. *et al. Science* **357**, 389–393 (2017).) and Pt-Mo₂N (Zhang, Z.-S. *et al. J. Am. Chem. Soc.* **142**, 13362–13371 (2020).) (Table R3).

Table R3. Comparison of catalytic performances for WGS reaction over various catalysts.

Catalyst	Reaction conditions	Reaction temperature (°C)	Reaction rate ($\mu\text{mol}_{\text{CO}} \text{g}_{\text{cat}}^{-1} \text{s}^{-1}$)	TOF (s^{-1})	Ref.
$\text{Ni}_9\text{Y}_1\text{O}_x$	2% CO, 10% H_2O , balance N_2	250	37.2	0.37	This work
		300	140.6	1.39	
		350	434.7	4.31	
1Re-10Ni/CeO ₂	5% CO, 10% H_2O , balance He	300	30.0	–	11
$\text{Ni}_{20}/\text{CeLaO}_x$	10% CO, 20 % H_2O , balance He	300	35.2	–	12
		350	36.3	–	
$\text{Cu}_4\text{Ni}_{16}/\text{CeLaO}_x$	10 % CO, 20 % H_2O , balance He	300	31.1	–	
		350	35.9	–	
$\text{Ni}@/\text{TiO}_{2-x}$ (450)	6% CO, 24% H_2O , 70% Ar	350	356.8	3.8	6
Inverse CeO ₂ /Cu	2% CO, 10% H_2O , balance N_2	300	47.5	0.058	1
Cu-Ce(La)O_x	2% CO, 10% H_2O , balance N_2	300	9.0	–	13
$\text{Cu}_{0.3}\text{Fe}_{0.7}\text{O}_x$	2% CO, 10% H_2O , balance N_2	300	12.2	–	14
Ir_1/FeO_x	2% CO, 10% H_2O , balance He	300	43.4	2.31	4
5Pt/0.06Na-TiO ₂	3% CO, 10% H_2O , balance He	250	38.4	1.58	5
2% Au/ α -MoC	10.5% CO, 21% H_2O , 20% N_2 , balance Ar	200	325	–	15
0.2Pt-Mo ₂ N	2% CO, 10% H_2O , balance N_2	300	232	–	16

Comment 2: On page 10, what exactly is the nature of the interaction between Ni/NiO and Y_2O_3 ?

Response: Thanks for the reviewer’s valuable comments and suggestions. The discussion about the nature of the interaction between Ni/NiO and Y_2O_3 was developed in two main aspects.

Firstly, the H_2 -TPR results shown that the actual H_2 consumption was higher than the theoretical value. This indicated that the Y_2O_3 species in the catalyst were also reduced due to

the formation of the interface. And this reduction was more expressed as the consumption of oxygen species. In other words, the interfacial interactions made the reduction of Y_2O_3 and favored the formation of a more active catalytic interface.

More comprehensively, **we performed DFT calculations to demonstrate the nature of the interaction was an electrostatic attraction.** Our previous investigation was not clear and definite enough (the structure of model was not changed), and so the description about the simulated model was optimized to $Y_3O_4/NiO_x/Ni\{111\}$. The top Ni-layer was separated from $Ni\{111\}$ and formed NiO_x -layer by adding four oxygen atoms. Three yttrium atoms along with the remainder oxygen atoms formed the Y_3O_4 layer. The deformation charge density diagram showed that the borderline between $Ni\{111\}$ and NiO_x was an accumulation area for electron, the cyan area in the Figure R6. The Bader charge analysis suggested that 0.187 negative charge transferred from $Ni\{111\}$ to NiO_x , i.e., there was no obvious gain and loss of electrons. Based these two aspects, we stood for the metallic bond. As regards Y_3O_4 , 1.643 negative charge transferred to NiO_x , and the bonding ways were Ni-O and Y-O. So, the chemical interaction between Y_3O_4 and NiO_x were the ionic bond.

The Figure R6 has been added as Supplementary Fig. 7 in the revised supporting information on page S11. The related data has been added in Supplementary Table 2 in the revised supporting information on page S23. The corresponding description has been shown from page 9, line 28 to page 10, line 3 in the revised manuscript and page S11, line 7–14 in the revised supporting information (highlighted in yellow).

Figure R6 The structural diagram of the $\text{Y}_3\text{O}_4/\text{NiO}_x/\text{Ni}\{111\}$ model: (a) exploded view; (b) (c) deformation charge density diagram (using VESTA¹⁷ software), the yellow and cyan represent the density of positive charge increases and decreases, respectively.

Comment 3: On pages 13 and 14, the model used in the DFT calculations needs a much better justification. What is the exact link between the model and the catalyst samples used to do the experiments?

Response: Thanks for the reviewer's valuable comments and suggestions. In the former vision of manuscript, the model named $\text{Y}_3\text{O}_8/\text{Ni}\{111\}$ did not come to the point, and the

model was given an optimized name, that was $\text{Y}_3\text{O}_4/\text{NiO}_x/\text{Ni}\{111\}$. We have divided the original model into three parts: Y_3O_4 , NiO_x and $\text{Ni}\{111\}$, and these matched well with the catalyst samples ($\text{Ni-NiO}_x\text{-Y}_2\text{O}_3$). The crystal face $\{111\}$ was the dominant exposed surface of Ni, so we chose $\text{Ni}\{111\}$ as the bases of the model. The NiO_x layer was derived from the partial oxidation of metallic nickel, and four oxygen atoms were loaded on the $\text{Ni}\{111\}$, shown as Figure R6a. As for Y_2O_3 , the chosen cluster contained three yttrium atoms and four oxygen atoms, and the Y_3O_4 cluster was derived from the unit cell of Y_2O_3 . Despite the non-stoichiometry, the Bader charge analysis showed that the net charge of Y atoms on the $\text{Y}_3\text{O}_4/\text{NiO}_x/\text{Ni}\{111\}$ was similar to those on the Y_2O_3 bulk, i.e., +1.95 e vs. +2.21 e. The positions of oxygen atom on the NiO_x layer were optimized to match the Y_3O_4 cluster, i.e., the position of the O atoms on the NiO_x could form chemical bonds with Y atoms on the Y_3O_4 cluster.

As for the rule of model building, our model was constructed with reference to the inverse catalyst with similarities to the catalyst we prepared. Rodriguez J. A. *et al.* investigated the WGS reaction by constructing small-sized TiO_{2-x} clusters on the Au (111) surface. Their model contains chains of TiO_2 over the gold substrate in a (3×1) array. They concluded that this model catalyst exposes not fully coordinated Ti centers, as expected for a TiO_2 NP, and allows the study of the oxide-metal interface. (Rodriguez J. A. *et al. Science* **318**, 1757–1760 (2007).) Rodriguez J. A. *et al.* also constructed the inverse $\text{CeO}_x/\text{Cu}(111)$ catalyst. Their small oxide particle model (triangle) of Ce_6O_{13} was constructed on the $\text{Cu}(111)$ surface based on the results of STM. Most of the O atoms in this model were shared by Ce and Cu, and the average oxidation state of Ce atoms was close to +3. Therefore, from an electronic viewpoint, the cerium cations in our model could be used to study the reactivity of the Ce^{3+} sites in small ceria particles ($\text{CeO}_x\text{-I}$) or in the interface between large ceria nanoparticles ($\text{CeO}_x\text{-II}$) and the copper substrate. (Rodriguez J. A. *et al. Angew. Chem. Int. Ed.* **48**, 8047–8050 (2009).) Yang F. *et al.* constructed representative CeO_2 (111) particles on Cu (111) based on STM results, and the most stable one in the reducing atmosphere was determined by optimization to be Ce_6O_{13} . (Yang F. *et al. J. Am. Chem. Soc.* **133**, 3444–3451 (2011).)

All of the above studied systems contain a large amount of metals on which oxide clusters were constructed. Our Ni₉Y₁O_x catalyst was also dominated by metallic Ni, and based on the HAADF-STEM results, it was believed that Y₂O₃ were mainly distributed in the surroundings of Ni particles with small size. And combined with the *in situ* Raman, we determined the presence of the Ni-NiO_x-Y₂O₃ interface. Based on these results, we placed Y₃O₄ clusters on the NiO_x/Ni{111} surface. Among them, Ni on the NiO_x was connected with O to simulate the interfacial structure, which could be used to study the contribution of the interface to catalysis. In order to give the readers a more direct understanding for the model building, we have changed the corresponding expression to “Y₃O₄/NiO_x/Ni{111} model” in the manuscript.

In addition, in order to show the rationality of the model construction more comprehensively, we have calculated the pathway of the WGS reaction catalyzed by this model based on the study of the H₂O dissociated, and the results were shown as Figure R7 and Table R5, 6, 7. The simulated reaction path was composed of eight elementary reactions. H₂O was adsorbed on the atom Y at the first step and CO was adsorbed on the atom Ni at third step. The rate-determining step was the fourth elementary reaction, i.e., *OH migration from atom Y to atom C, and the most probable cause was that the adsorption energies of *OH and CO were too strong. As a whole, the WGS reaction was spontaneous because the sign of $\Delta_r G_m$ was negative (-2.617 eV near the gas inlet, and -1.476 eV near the gas outlet at 300 °C). Besides, we also corrected the partial pressure of H₂O for the H₂O adsorption and dissociation processes. The related results have been modified in the manuscript. **The Figure R7 has been divided into Fig. 5d in the revised manuscript on page 15 and Supplementary Fig. 17 in the revised supporting information on page S21. The Table R5, 6, 7 has been added as Supplementary Table 6, 7, 8 in the revised supporting information on page S27–29. And the corresponding description has been shown from page 14, line 4 to page 15, line 7 in the revised manuscript (highlighted in yellow).**

Figure R7 The simulated WGS reaction pathways (the reaction temperature: 300 °C, outlet pressure of the reactor: the partial pressure of H₂O, CO, CO₂ and H₂ were listed in Table R5; inlet pressure of the reactor: the partial pressures of CO, H₂O, CO₂ and H₂ were 2000, 10000, 1 and 1 Pa respectively.)

Table R5 The partial pressures of CO, H₂O, CO₂ and H₂ at air outlet (unit Pa).

Temperature (°C)	partial pressures (Pa)			
	CO	H ₂ O	CO ₂	H ₂
150	2000	10000	1	1
200	1938	9938	61	61
212	1680	9680	319	319
223	1444	9444	555	555
235	1034	9034	966	966
247	408	8408	1592	1592
256	101	8101	1899	1899
300	1	8000	2000	2000
350	1	8000	2000	2000

* One Pa in the table means the substance is not detected.

Table R6 Cumulative changes in the Gibbs free energies (ΔG) of the WGS reaction at air outlet (The partial pressures of CO, H₂O, CO₂ and H₂ are listed in Supplementary Table 7).

Temperature (°C)	ΔG (eV)							
	IMI-1	IMI-2	IMI-3	IMI-4	IMI-5	IMI-6	IMI-7	Surface
150	-0.521	-0.723	-2.282	-1.279	-1.519	-1.241	-2.031	-1.835
200	-0.489	-0.707	-2.173	-1.168	-1.404	-1.158	-2.256	-2.054
212	-0.481	-0.703	-2.141	-1.136	-1.37	-1.133	-2.163	-1.927
223	-0.473	-0.698	-2.111	-1.105	-1.338	-1.109	-2.115	-1.886
235	-0.463	-0.693	-2.069	-1.063	-1.295	-1.074	-2.054	-1.834
247	-0.453	-0.687	-1.998	-0.992	-1.223	-1.011	-1.966	-1.756
256	-0.445	-0.682	-1.913	-0.905	-1.136	-0.932	-1.876	-1.681
300	-0.417	-0.67	-1.584	-0.576	-0.803	-0.633	-1.555	-1.476
350	-0.385	-0.659	-1.455	-0.444	-0.666	-0.54	-1.432	-1.487

Table R7 Cumulative changes in the Gibbs free energies (ΔG) of the WGS reaction at air inlet, i.e., partial pressures of CO, H₂O, CO₂ and H₂ are 2000, 10000, 1 and 1 Pa respectively.

Temperature (°C)	ΔG (eV)							
	IMI-1	IMI-2	IMI-3	IMI-4	IMI-5	IMI-6	IMI-7	Surface
150	-0.521	-0.723	-2.282	-1.279	-1.519	-1.241	-2.031	-1.835
200	-0.490	-0.707	-2.175	-1.170	-1.406	-1.160	-2.426	-2.393
212	-0.482	-0.704	-2.150	-1.144	-1.379	-1.141	-2.413	-2.419
223	-0.475	-0.701	-2.127	-1.121	-1.355	-1.125	-2.402	-2.444
235	-0.468	-0.698	-2.102	-1.096	-1.329	-1.108	-2.390	-2.471
247	-0.460	-0.694	-2.078	-1.071	-1.303	-1.091	-2.377	-2.498
256	-0.455	-0.692	-2.060	-1.052	-1.282	-1.078	-2.367	-2.519
300	-0.428	-0.681	-1.972	-0.964	-1.190	-1.020	-2.319	-2.617
350	-0.398	-0.671	-1.876	-0.866	-1.088	-0.961	-2.263	-2.727

Comment 4: Top of page 15, OK the addition of Y₂O₃ enhances the catalytic performance of Ni for the WGS reaction. Why that finding should be published in top-level journal like Nature Communications? It looks like a nice finding for a regular journal in catalysis.

Response: Thanks to the reviewer for the high evaluation of the manuscript. This manuscript provides new concept of interface design, and put a range of early transition fifth and sixth row metal oxides on to the screening radar for catalysis research. The ability to activate thermodynamically stable molecules such as H₂O and CO₂ is the key innovation.

First of all, the construction of catalytically active interfaces between metals and supports has attracted a lot of attention. It is crucial to design and control suitable metal-support interactions to create abundant and effective interfacial active sites for molecular transformations. While interfaces formed by many third row and fourth row 3d metal oxides are more intensively studied so far (Wang, H. *et al. Nat. Catal.* **4**, 418–424 (2021); Matsubu, J. C. *et al. Nat. Chem.* **9**, 120–127 (2017); Herzing, A. A. *et al. Science* **321**, 1328–1331 (2008), and so on.), **the construction and application of metal-rare earth oxides (except for CeO₂) interfaces are not yet mature.** These early transition rare earth cations are strong Lewis acids and have strong binding to Lewis bases such as H₂O and NH₃. Therefore, **obtaining and understanding the structure and role of such interfaces is important for environmental catalysis, energy catalysis, and expanding the applications of rare earth materials.**

In addition, the water gas shift (WGS) reaction usually considers H₂O dissociation as the rate-determining step, which is beneficial for us to investigate the catalytic properties of the interface as a research object. For Ni-based catalysts, current reports show that they are mainly used to catalyze high-temperature (HT) WGS reactions (Xu, M. *et al. J. Am. Chem. Soc.* **140**, 11241–11251 (2018); Ang, M. L. *et al. ACS Catal.* **4**, 3237–3248 (2014); Saw, E. T. *et al. J. Catal.* **314**, 32–46 (2014), and so on.). Decreasing application temperature of Ni-based catalysts to the medium-temperature (MT) region can achieve rapid kinetics with high CO conversion equilibrium. Therefore, **it is significant to prepare a composite rare earth Ni-based catalyst for MT-WGS reaction which contains an efficient catalytic interface. Moreover, the dynamic change and catalytic effect of such interface during the reaction is needed to clear understanding.**

In this manuscript, we have successfully constructed sufficient and stable Ni-NiO_x-Y₂O₃ interfacial sites, which exhibit excellent catalytic performance for MT-WGS reaction. **We believe our findings will interest a broad readership in catalysis and materials science according to the following aspects:**

- 1. We have successfully prepared Ni-Y₂O₃ composite catalyst and transferred the WGS reaction operating temperature from HT to MT for Ni-based catalyst.** The reaction

rate of the Ni-Y₂O₃ catalyst with Ni-NiO_x-Y₂O₃ interfacial structure reached 140.6 μmol_{CO} g_{cat}⁻¹ s⁻¹ at 300 °C, achieving the highest activity of Ni-based catalysts. Besides, it exhibited solid stability under harsh reaction conditions (300 °C, space velocity of 420,000 cm³ g_{cat}⁻¹ h⁻¹).

2. We created the catalytically efficient Ni-NiO_x-Y₂O₃ interface sites in the Ni-Y₂O₃ catalyst and it was preserved during the WGS reaction process. We clearly observed the formation of Ni-Y₂O₃ interface, and the dynamic structural change of the Ni-NiO_x-Y₂O₃ interface was clearly recorded by *in situ* Raman. These findings indicated that our understanding of the interface was deep enough. The real active interface structure in the WGS reaction catalyzed by Ni-based catalysts was finely specified.

3. We demonstrated the realistic role of the Ni-NiO_x-Y₂O₃ interfaces during the catalytic process. Based on the experimental investigation and DFT calculation, the Ni-NiO_x-Y₂O₃ interfaces effectively dissociated H₂O molecules and facilitated the reaction in accordance with the associative mechanism. Thus, the significance of the Ni-NiO_x-Y₂O₃ interface for catalytic performance, as well as the reaction mechanism was adequately proved.

After all, this manuscript focuses on the construction, dynamic change and catalysis of the Ni-NiO_x-Y₂O₃ interface, which represents an important academic breakthrough **in the fields of catalysis, materials and surface research. We think that it can meet the scope of your journal, and is justified for the *Nature Communications*.** We are looking forward to your next comments.

References

1. Yan, H. et al. Construction of stabilized bulk-nano interfaces for highly promoted inverse CeO₂/Cu catalyst. *Nat. Commun.* **10**, 3470 (2019).
2. Fu, X. et al. Direct identification of active surface species for the water-gas shift reaction on a gold-ceria catalyst. *J. Am. Chem. Soc.* **141**, 4613–4623 (2019).
3. Schilling, C. and Hess C. Elucidating the role of support oxygen in the water–gas shift reaction over ceria-supported gold catalysts using operando spectroscopy. *ACS Catal.* **9**, 1159–1171 (2019).
4. Lin, J. et al. Remarkable performance of Ir₁/FeO_x single-atom catalyst in water gas shift reaction. *J. Am. Chem. Soc.* **135**, 15314–15317 (2013).
5. Panagiotopoulou, P., Kondarides, D. I. Effects of alkali promotion of TiO₂ on the chemisorptive properties and water–gas shift activity of supported noble metal catalysts. *J. Catal.* **267**, 57–66 (2009).
6. Xu, M. et al. TiO_{2-x}-modified Ni nanocatalyst with tunable metal–support interaction for water–gas shift reaction. *ACS Catal.* **7**, 7600–7609 (2017).
7. Xu, M. et al. Insights into interfacial synergistic catalysis over Ni@TiO_{2-x} catalyst toward water–gas shift reaction. *J. Am. Chem. Soc.* **140**, 11241–11251 (2018).
8. Ang, M. L. et al. Highly active Ni/xNa/CeO₂ catalyst for the water–gas shift reaction: Effect of sodium on methane suppression. *ACS Catal.* **4**, 3237–3248 (2014).
9. Ang, M. L. et al. High-temperature water–gas shift reaction over Ni/xK/CeO₂ catalysts: Suppression of methanation via formation of bridging carbonyls. *J. Catal.* **329**, 130–143 (2015).
10. Saw, E. T. et al. Bimetallic Ni–Cu catalyst supported on CeO₂ for high-temperature water–gas shift reaction: Methane suppression via enhanced CO adsorption. *J. Catal.* **314**, 32–46 (2014).
11. Chayakul, K., Srithanratana T. and Hengrasmee S. Catalytic activities of Re–Ni/CeO₂ bimetallic catalysts for water gas shift reaction. *Catal.Today* **175**, 420–429 (2011).
12. Lin, J. et al. Hydrogen production by water–gas shift reaction over bimetallic Cu–Ni catalysts supported on La-doped mesoporous ceria. *Appl. Catal. A: Gen.* **387**, 87–94 (2010).
13. Li, Y., Fu, Q. & Flytzani-Stephanopoulos, M. Low-temperature water-gas shift reaction over Cu- and Ni-loaded cerium oxide catalysts. *Appl. Catal. B* **27**, 179–191 (2000).
14. Yan, H. et al. Promoted Cu-Fe₃O₄ catalysts for low-temperature water gas shift reaction: optimization

- of Cu content. *Appl. Catal. B* **226**, 182–193 (2018).
15. Yao, S. et al. Atomic-layered Au clusters on α -MoC as catalysts for the low-temperature water-gas shift reaction. *Science* **357**, 389–393 (2017).
 16. Zhang, Z.-S. et al. Intrinsically active surface in a Pt/ γ -Mo₂N catalyst for the water–gas shift reaction: molybdenum nitride or molybdenum oxide? *J. Am. Chem. Soc.* **142**, 13362–13371 (2020).
 17. Momma, K. and Izumi, F. VESTA 3 for three-dimensional visualization of crystal, volumetric and morphology data. *J. Appl. Crystallogr.* **44**, 1272–1276 (2011).

REVIEWERS' COMMENTS

Reviewer #1 (Remarks to the Author):

Authors have adequately taken into account the reviewer's comments, and have addressed their recommendations.

The paper can be now accepted for publication.

Reviewer #3 (Remarks to the Author):

The authors have addressed my previous comments and now they have delivered a much better article. Thus, I recommend acceptance for publication.

Responses to the Reviewers' Comments and the Corresponding Revisions

Reviewer #1:

Comment: Authors have adequately taken into account the reviewer's comments, and have addressed their recommendations. The paper can be now accepted for publication.

Response: Thanks for reviewer's comments. After careful consideration of the reviewers' comments and supplemented the manuscript, which greatly improved the quality of our research work. Thanks for the reviewer again.

Reviewer #3:

Comment: The authors have addressed my previous comments and now they have delivered a much better article. Thus, I recommend acceptance for publication.

Response: Thanks for reviewer's comments. The professional and detailed comments of the reviewers were very helpful for us, the quality of our manuscript has been greatly improved compared to the original version. Thanks for the reviewer again.